# Specific binding of GPR174 by endogenous lysophosphatidylserine leads to high constitutive G_s signaling

Yingying Nie[1,2,6], Zeming Qiu[2,3,6], Sijia Chen[2], Zhao Chen[2,3], Xiaocui Song[2], Yan Ma[2,3], Niu Huang[2,3], Jason G. Cyster[4,5] & Sanduo Zheng[1,2,3]✉

Many orphan G protein-coupled receptors (GPCRs) remain understudied because their endogenous ligands are unknown. Here, we show that a group of class A/rhodopsin-like orphan GPCRs including GPR61, GPR161 and GPR174 increase the cAMP level similarly to fully activated D1 dopamine receptor (D1R). We report cryo-electron microscopy structures of the GPR61–G_s, GPR161–G_s and GPR174–G_s complexes without any exogenous ligands. The GPR174 structure reveals that endogenous lysophosphatidylserine (lysoPS) is copurified. While GPR174 fails to respond to exogenous lysoPS, likely owing to its maximal activation by the endogenous ligand, GPR174 mutants with lower ligand binding affinities can be specifically activated by lysoPS but not other lipids, in a dose-dependent manner. Moreover, GPR174 adopts a non-canonical G_s coupling mode. The structures of GPR161 and GPR61 reveal that the second extracellular loop (ECL2) penetrates into the orthosteric pocket, possibly contributing to constitutive activity. Our work definitively confirms lysoPS as an endogenous GPR174 ligand and suggests that high constitutive activity of some orphan GPCRs could be accounted for by their having naturally abundant ligands.

Given their wide distribution in the human body, versatile ligand recognition, and diverse downstream pathways, GPCRs contribute to virtually every aspect of physiology and pathology. As such, GPCRs are the most important class of targets for drug discovery campaigns. GPCRs are known to recognize a wide range of stimuli including light, odors, neurotransmitters, hormones, and peptides. However, there are over 100 GPCRs with no characterized endogenous ligands and clear biological functions, which are called orphan GPCRs (oGPCRs)[1,2]. Defining the endogenous ligands for oGPCRs is very important for understanding their physiological functions, and opens new possibilities for drug discovery against diseases with unmet medical needs. Notably, a large number of GPCRs including some oGPCRs show constitutive activity[3,4], isomerizing from inactive state to active state in the absence of exogenous agonists. The discovery of engineered adrenoceptors mutants and disease-causing GPCR mutants with increased constitutive activity suggested that the constitutive activity of GPCRs is partly attributed to their intrinsic molecular properties[3,5]. For instance, an 'ionic lock' between the conserved D/ERY motif on TM3 is required to maintain the inactive state. Mutations that disrupt the ionic lock lead to receptor activation and increase constitutive activity[6,7]. Moreover, recent structural studies have shown that some oGPCRs that exhibit high constitutive activity may be self-activated by the second extracellular loop (ECL2), which penetrates into the orthosteric binding pocket (OBP)[8–11].

Building upon previous studies that evaluated the constitutive activity of 40 class A oGPCRs using a cAMP-dependent response

[1]College of Life Sciences, Beijing Normal University, 100875 Beijing, China. [2]National Institute of Biological Sciences, 102206 Beijing, China. [3]Tsinghua Institute of Multidisciplinary Biomedical Research, Tsinghua University, 100084 Beijing, China. [4]HHMI, University of California, San Francisco, CA 94143, USA. [5]Department of Microbiology and Immunology, University of California, San Francisco, CA 94143, USA. [6]These authors contributed equally: Yingying Nie, Zeming Qiu. ✉e-mail: zhengsanduo@nibs.ac.cn

element (CRE) based reporter assay[12], we comprehensively analyzed the constitutive activity of 81 class A oGPCRs (annotated in GPCRdb[13]) expressed in Expi293F cells using the GloSensor cAMP accumulation assay. We found that 12 oGPCRs remarkably increased the intracellular cAMP level, compared with other oGPCRs and D1 dopamine receptors (D1R) without ligand treatment (Fig. 1a and Supplementary Table 1). Most of these receptors are expressed at lower levels compared to D1R (Supplementary Fig. 1a). Further efforts to increase the expression level of D1R have little effect on its basal activity, which is still remarkably lower than that of GPR174 even when expressed at a low level (Supplementary Fig. 1b), suggesting that the higher constitutive activities of these receptors are not attributed to their expression levels. These constitutively active oGPCRs are abbreviated as caoGPCRs hereafter. Interestingly, most of the caoGPCRs are closely related to GPCRs known to be activated by lipids (Supplementary Fig. 1c). Based on their sequence and structural similarity, they are further classified into three major groups: I (GPR26, GPR78, GPR101, GPR161), II (GPR3, GPR6, GPR12), III (GPR21, GPR52). The receptors in group I are closely related to prostanoid receptors. Structural predictions by Alphafold reveal that ECL2 in group I form an antiparallel β-sheet, covering the ligand-binding pocket, which is observed in prostanoid receptors (Supplementary Fig. 1d). A phytoestrogen, genistein has been reported to increase the cAMP level in cells expressing GPR26 and GPR78 (US patents: US7872101B1). GPR3, GPR6, and GPR12 in group II share about 60% sequence identity, and are phylogenetically related to sphingosine-1-phosphate receptors (S1PR) and cannabinoid receptors (Supplementary Fig. 1c). S1P, sphingosylphosphorylcholine (SPC) and cannabidiol have been identified as putative modulators of GPR3, GPR6, and GPR12[14]. GPR21 and GPR52 with 71% sequence identity in group III share about 30% sequence similarity with GPR119 that can be activated by lysophosphatidylcholine (lysoPC)[15,16]. While the endogenous ligands of GPR21 and GPR52 have not been identified, several surrogate agonists for GPR52 show high lipophilicity and poor water solubility, suggesting the hydrophobicity of the ligand binding pocket. Recent structural studies suggested that GPR21 and GPR52 can be self-activated by ECL2, which contributes to their high constitutive activity. GPR174 has been shown to be activated by lysophophatidylserine (lysoPS) via Gα12/13 and Gαs[4,17,18]. LysoPS-mediated GPR174 activation restrains T regulatory cell development and function and conventional

T cell proliferation and modulates B cell gene expression via Gαs[19–23]. Given the possible relationship between lipids and caoGPCRs, we reasoned that the high constitutive activity of caoGPCRs may be attributed to the stimulation by endogenous lipids released from transfected cells where they are expressed. Therefore, we sought to determine the cryo-electron microscopy (cryo-EM) structures of caoGPCR–Gs signaling complexes purified from mammalian cells to unravel molecular mechanisms underlying constitutive activity.

In this work, we show that the high constitutive activity of GPR174 can be attributed to endogenous lysoPS, which occupies the OBP of GPR174, whereas penetration of ECL2 in the OBP of GPR61 and GPR161 possibly contributes to the high constitutive activity. Some Gs-coupled receptors including GPR174 adopt a non-canonical Gs coupling mode, where TM6 shows less pronounced movement compared with that in Gi/o-coupled receptors. We found that the non-canonical Gs coupling mode is due to the presence of a larger hydrophobic residue (L or C) at position 5.65 of receptors, while a small hydrophobic residue (A or V) at this position leads to the canonical Gs coupling mode.

## Results

### Structural approach to identify endogenous ligands of oGPCRs

In addition to increasing cAMP levels, caoGPCRs recruit Gαs more efficiently than most other oGPCRs (Fig. 1a and Supplementary Table 1), as shown by NanoBiT mini-Gαs recruitment assay[24]. Moreover, they all contain a large hydrophobic residue (L, M, or F) at position 34.51 of the second intracellular loop (ICL2) (Fig. 1b), which is essential for Gs coupling[25]. These results indicate that caoGPCRs can increase the cAMP level through Gs. Notably, the cAMP levels induced by expression of GPR174 alone are almost two-fold of those induced by fully activated D1R treated with dopamine (Fig. 1c). Although previous studies have shown that lysoPS can activate GPR174 in a dose-dependent manner using a transforming growth factor-α (TGFα) shedding assay[17], our GloSensor cAMP assay results revealed that lysoPS failed to further increase cAMP levels in cells expressing GPR174 (Fig. 1c). Owing to its amphipathic characteristics, very high concentration of lysoPS disrupts membrane structure and causes cell lysis, thereby leading to a reduced cAMP level. We speculated the endogenous ligands might have occupied the receptor, leading to its maximal activation in the cAMP assay. To test this hypothesis, we

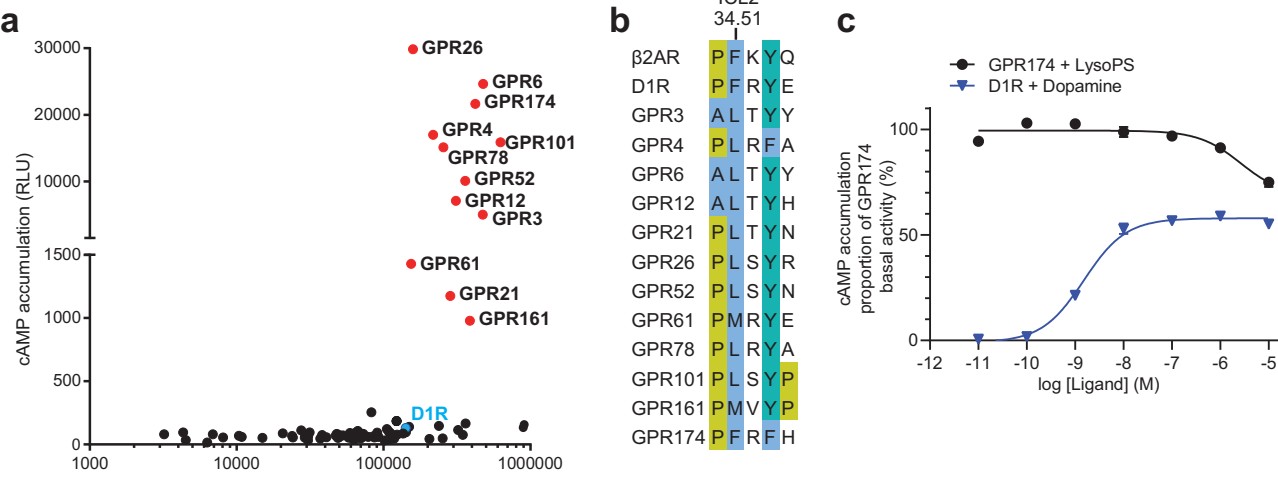

**Fig. 1 | A group of class A oGPCRs dramatically increase cAMP levels via the Gs pathway. a** GloSensor cAMP accumulation assay and NanoBiT mini-Gs recruitment assay performed in Expi293F cells transiently expressing any of 81 oGPCRs or D1R with relative luminescence unit (RLU) value shown. Receptors that show high cAMP levels compared with the other oGPCRs are colored in red. **b** Sequence alignment of ICL2 in caoGPCRs, D1R and β2AR. Residues are highlighted based on the Clustal color scheme. **c** Concentration–response curves in the cAMP accumulation assay at Expi293F cells transiently transfected with GPR174 or D1R treated with lysoPS or dopamine. Each data point represents mean ± SEM from three independent experiments. Source data are provided as a Source Data file.

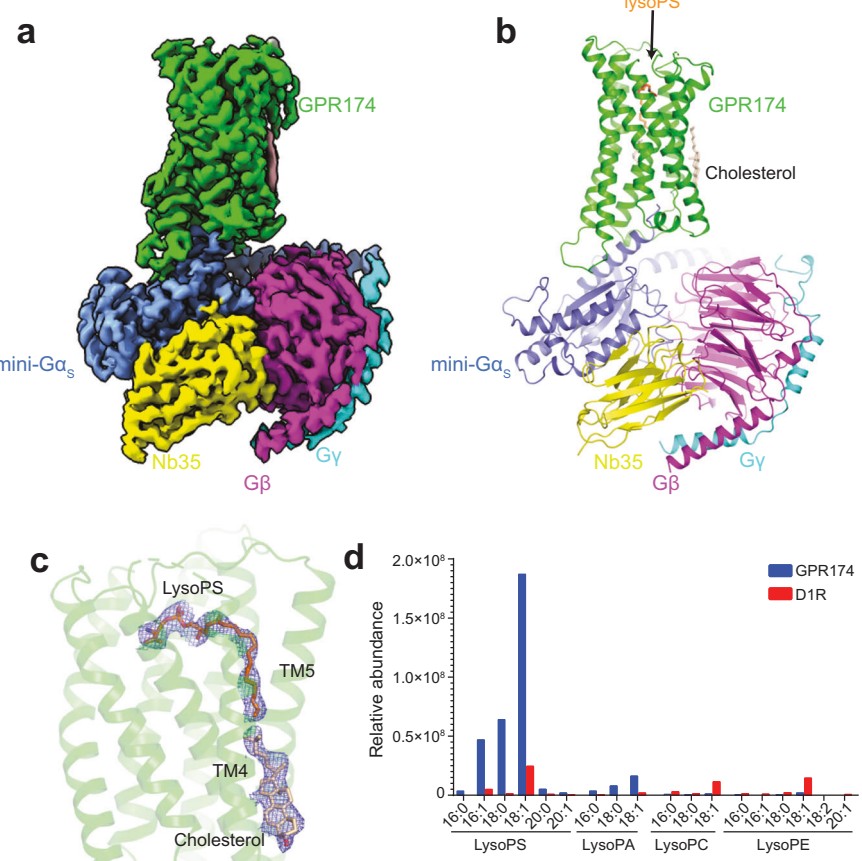

**Fig. 2 | The endogenous lysoPS is copurified with GPR174. a** Cryo-EM map of the GPR174 and mini-G$_s$ complex. **b** Ribbon representation of the GPR174 and mini-G$_s$ complex. LysoPS and cholesterol are shown as sticks. **c** Cryo-EM density map of lysoPS and cholesterol in the GPR174–G$_s$ complex at a contour level of 3.5σ.

**d** Lipidomics analysis of the GPR174–G$_s$ and D1R–G$_s$ complexes by MS. The relative abundance of representative LPs with the acyl chain length from 16:0 to 20:1 determined by MS is shown. Source data are provided as a Source Data file.

attempted to identify the endogenous ligands of caoGPCRs using structural approaches complemented by mass spectrometry (MS). Since G protein binding with GPCRs can enhance agonist binding[26], we purified GPR174–mini-Gα$_s$ fusion protein from Expi293F cells (the same cell type used for our signaling assays) through affinity purification and assembled GPR174–mini-Gα$_s$ fusion protein with purified Gβ$_1$γ$_2$ and Nb35 using size-exclusion chromatography in the absence of exogenous ligands (Supplementary Fig. 2a). The structure of the GPR174–G$_s$ complex was determined using cryo-EM to a nominal resolution of 2.9 Å (Supplementary Fig. 2b–f and Supplementary Table 2). An unassigned density was observed in the orthosteric pocket of GPR174 after building all protein structure models (Fig. 2a–c). LysoPS with 15 carbons in the acyl chain can be precisely modeled due to the high quality of EM density. To further confirm the identity of lysoPS, we performed lipidomics analysis of GPR174–G$_s$, using D1R–G$_s$ as a control, by MS (Fig. 2d and Supplementary 3a). LysoPS isoforms with acyl chain lengths ranging from 16:0 to 20:1 were remarkably more abundantly present in GPR174–G$_s$, compared with D1R–G$_s$. As 18:1 lysoPS is the most dominant lipid present in GPR174, we further verified its binding pose using molecular dynamics (MD) simulations. LysoPS stably associates with GPR174 during the course of 100-ns MD (Supplementary Fig. 3b). The acyl chain particularly the first two carbons displayed more flexibility than the polar group, explaining the absence of the terminal carbons in the structure. Noteworthily, lysophospholipids (LPs) present in protein samples are mainly synthesized by cells as they are barely detected in the fresh culture medium. Although cells release a larger quantity of lysoPE and lysoPC than lysoPS[27], lysoPS is specifically copurified with GPR174 after a harsh

purification procedure, indicating their high affinity and specific binding. Since lysoPS can be produced by many and possibly all mammalian cells, it is not surprising that GPR174 also exhibits high basal activity when expressed in other cell lines such as Chinese Hamster Ovary (CHO) cells and HeLa cells (Supplementary Fig. 3c, d).

## LysoPS recognition by GPR174

When bound to GPR174, lysoPS forms an L-shape configuration. Its polar head group lies in the orthosteric pocket, parallel to the membrane plane, whereas the fatty acyl chain is nearly perpendicular to the membrane and extends into the membrane portal between TM4 and TM5 (Fig. 3a, b). LysoPS shows perfect charge and shape complementarity with the binding pocket of the receptor and buries a surface area of 614 Å$^2$, accounting for its high-affinity binding with GPR174. Notably, underneath lysoPS is a cholesterol molecule that binds on the surface of TM3, TM4, and TM5. Because of their close contact, cholesterol may potentially further enhance lysoPS binding to the receptor, suggesting a positive allosteric effect of the endogenous cholesterol (Fig. 2c). Cholesterol is also observed at a similar position in D1R, and interacts with a positive allosteric modulator LY3154207[28]. Specifically, the terminal amine group of serine in lysoPS engages in a hydrogen bond and a cation–π interaction with Y79[2.64] and Y22[1.35], respectively, while it's carboxyl acid group forms salt bridge interactions with R75[2.60] (Fig. 3c and Supplementary Fig. 2f). The phosphate group is covered by three spatially separated basic residues, namely R18[1.31], R156[4.64], and K257[6.62] (Fig. 3c). Y99[3.33] underneath lysoPS forms hydrogen bonds with hydroxyl groups at the sn-2 and sn-3 positions of the glycerol moiety. The acyl chain is encircled by a number of

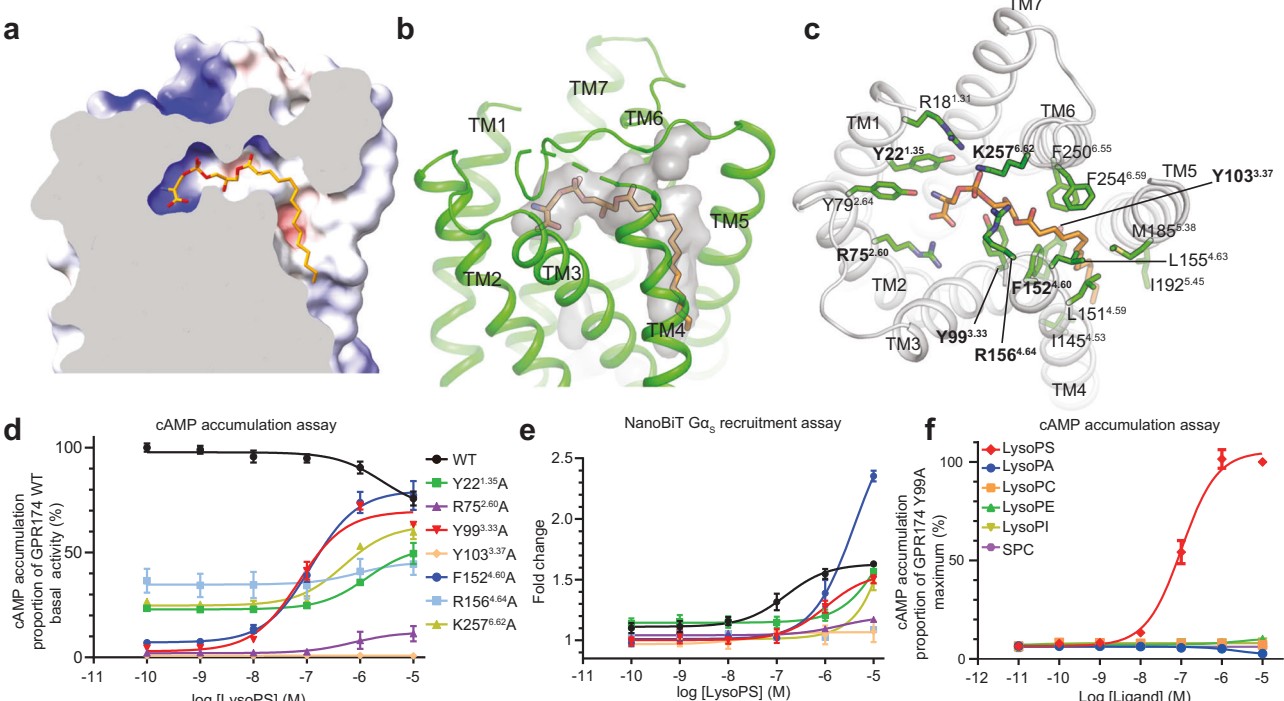

**Fig. 3 | Specific recognition of lysoPS by GPR174. a** Electrostatic potential of the lysoPS binding surface colored from red to blue for negatively and positively charged regions. **b** GPR174 shows perfect shape complementarity with lysoPS. The surface of interior pocket of GPR174 is shown in gray. **c** Detailed interactions between GPR174 and lysoPS. **d** and **e** Concentration–response curves of GPR174 mutants treated with increasing concentrations of lysoPS in the cAMP accumulation assay (**d**) and the NanoBiT Gαs recruitment assay (**e**). All measurements were performed in three independent times, shown as mean ± SEM. Source data are provided as a Source Data file. **f** Concentration–response curves of GPR174/Y99$^{3.33}$A mutants treated with different LPs in the cAMP accumulation assay. Each data point represents mean ± SEM from three independent experiments. Source data are provided as a Source Data file.

hydrophobic residues including Y103$^{3.37}$, L151$^{4.59}$, F152$^{4.60}$, L155$^{4.63}$, M185$^{5.38}$, F250$^{6.55}$, and F254$^{6.59}$. Consistent with our structural observations, mutation of R75$^{2.60}$, Y99$^{3.33}$, Y103$^{3.37}$, and F152$^{4.60}$ almost completely suppressed the basal activity; mutation of Y22$^{1.35}$, R156$^{4.64}$, or K257$^{6.62}$ reduced the basal activity by more than 50% (Supplementary Fig. 3e). Interestingly, in contrast to GPR174 wild-type (WT), most mutants can be activated by lysoPS in a dose-dependent manner with distinct potency (EC$_{50}$) and maximal effect (E$_{max}$) (Fig. 3d). LysoPS can activate the Y99$^{3.33}$A and F152$^{4.60}$A mutants with E$_{max}$ nearly approaching the basal activity of WT and potencies of 71.6 and 119.7 nM, respectively (Supplementary Table 3). Given the lack of potency of lysoPS for GPR174 WT in the cAMP assay, it is unclear to what extent these mutations alter the potency of lysoPS. Therefore, we sought to measure the potency of lysoPS for WT using the NanoBiT G$_s$ recruitment assay which has a lower amplification than the cAMP assay (Fig. 3e). For D1R, the potencies of dopamine obtained from the NanoBiT assay and cAMP assay are 2.1 µM and 3.6 nM (about 1000-fold difference), respectively (Supplementary Fig. 3f, g). As expected, lysoPS can activate GPR174 WT in a dose-dependent manner with a potency of 155.7 nM using the NanoBiT G$_s$ recruitment assay (Supplementary Table 3), and all mutations of GPR174 tested above remarkably reduce the potency of lysoPS, which is consistent with results using the cAMP assay (Fig. 3d and e and Supplementary Fig. 3e). For the Y99$^{3.33}$A mutant, the potency is reduced by about 7-fold compared to GPR174 WT in the NanoBiT assay (Supplementary Table 3). Therefore, we speculate that the potency of lysoPS for WT using the cAMP assay could be in a single-digit nanomolar range. The concentration of lysoPS generated by cells is likely to be higher than that required to produce the maximal effect of WT, explaining how GPR174 exhibits extremely high basal activity and that the exogenous lysoPS cannot further increase the cAMP level. The potency of the Y99$^{3.33}$A mutant

using the cAMP assay is comparable to that of WT using the TGFα shedding assay, which exceeds the concentration of lysoPS released by Expi293F or HEK293 cells explaining that they show dose–response curves when treated with exogenous lysoPS. Although GPR119 can be activated by lysoPC and is more abundantly synthesized than lysoPS in cells[15,16], its basal activity is similar to D1R without ligand treatment because of the low micromolar potency of lysoPC for GPR119 (Supplementary Table 1). Moreover, we tested activity of other LPs for the Y99$^{3.33}$A mutant, and none of them could activate this mutant, supporting the specific recognition of lysoPS by GPR174 (Fig. 3f). From a structural perspective, lysoPE, lysoPC and lysophosphatidic acid (lysoPA) that lack the terminal carboxylic acid group involved in binding R75$^{2.60}$ is analogous to R75$^{2.60}$ mutation which almost abolished the activity of lysoPS (Fig. 3d and e and Supplementary Fig. 3a). The inositol group in lysophosphatidylinositol (lysoPI) that is larger than serine in lysoPS cannot be accommodated in the binding pocket of GPR174 (Fig. 3b).

## Occupation of ECL2 in the orthosteric pocket of GPR161 and GPR61

Recently published structures of GPR12[11] in subgroup II and GPR21[9,10] and GPR52[8] in subgroup III in complex with G$_s$ revealed no ligands bound to the receptors, but instead showed that ECL2 of these receptors penetrate into the orthosteric pocket, leading to a self-activation model. Since no structures of receptors in group I have been reported, we further determined the cryo-EM structures of GPR161–G$_s$ and GPR61–G$_s$ complexes using the same strategy as GPR174 at nominal resolutions of 3.1 and 3.2 Å, respectively (Supplementary Figs. 4, 5 and Supplementary Table 2). The constitutively active GPR161 localizes to cilia and inhibits Sonic hedgehog signaling (Shh) by proteolysis of full-length Gli3 into its repressor form via the G$_s$-cAMP-protein kinase A

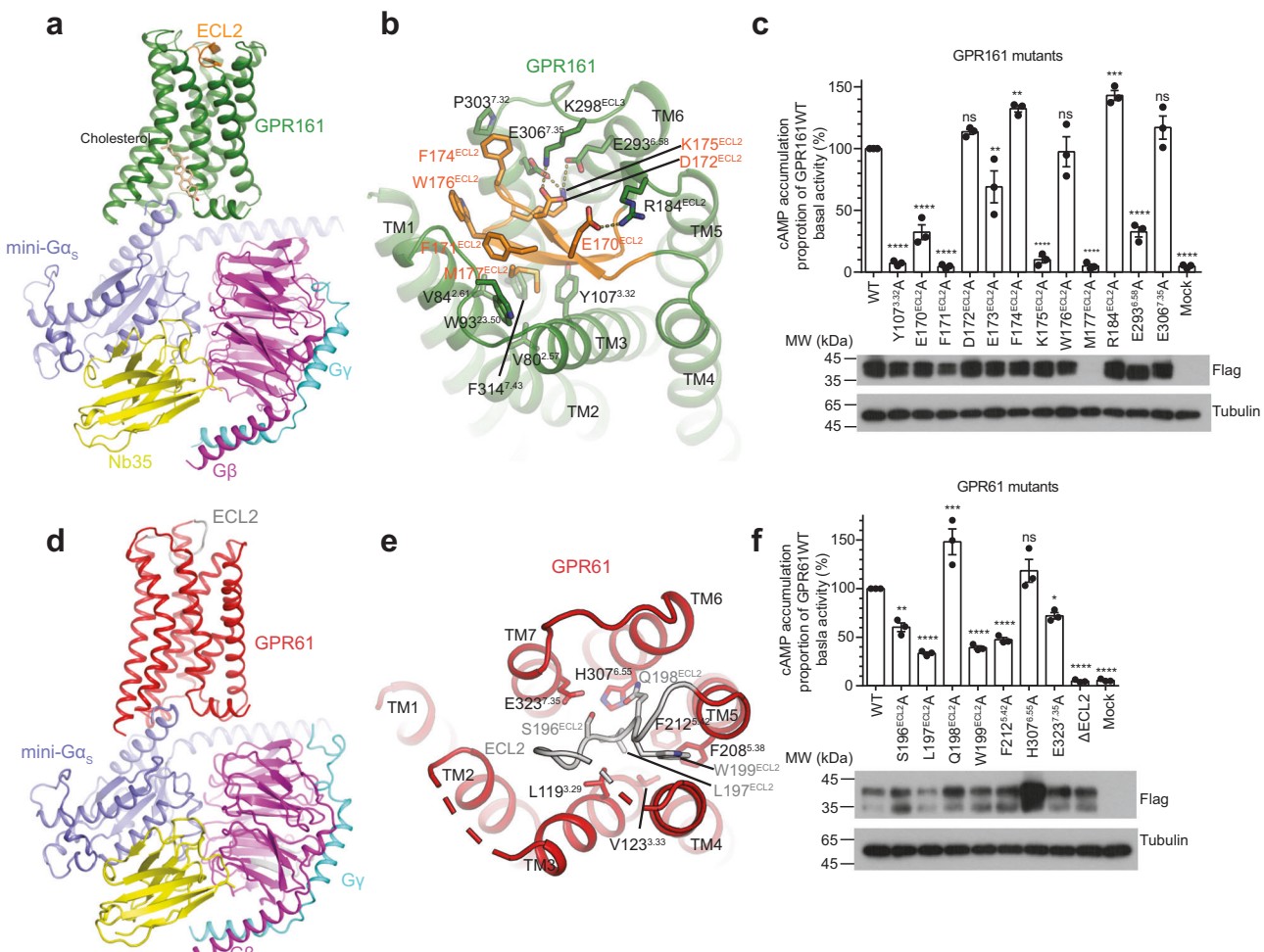

**Fig. 4 | ECL2 is penetrated into the orthosteric binding pocket of GPR161 and GPR61. a** and **d** Ribbon representation of the GPR161–G$_s$ (**a**) and GPR61–G$_s$ (**d**) complexes with ECL2 colored in orange and gray respectively. **b** and **e** Detailed interactions between ECL2 and other regions of GPR161 (**b**) or GPR61 (**e**). **c** and **f** The basal activity of GPR161 (**c**) or GPR61 (**f**) mutants determined by the cAMP assay. Bar graphs represent mean ± SEM from three independent experiments. Expression levels of mutants were determined by western blot using an anti-Flag antibody.

Statistical analysis was performed using one-way ANOVA method (ns, not significant; *$P < 0.1$; **$P < 0.01$; ***$P < 0.001$; ****$P < 0.0001$). Source data are provided as a Source Data file. $P$ values for GPR161 mutants versus WT are <0.0001, <0.0001, <0.0001, 0.5954, 0.0098, 0.0067, <0.0001, 0.9996, <0.0001, 0.0003, <0.0001, 0.3316, <0.0001 (from left to right). $P$ values for GPR61 mutants versus WT are 0.001, <0.0001, 0.0001, <0.0001, <0.0001, 0.2203, 0.0230, <0.0001, <0.0001 (from left to right).

pathway[29–31], which is required for normal embryonic development. GPR161 has been shown to be associated with medulloblastoma and breast cancer[32–34]. The GPR161 structure reveals that ECL2 forms an anti-parallel β sheet and is buried inside the receptor (Fig. 4a). The β-sheet structure of ECL2 universally exists in GPCRs that recognize prostanoids and peptides[35]. In the absence of ligands, ECL2 in peptide GPCRs is flexible and solvent-accessible and adopts distinct conformations when bound to different ligands. In contrast, reminiscent of prostanoid receptors, one β strand of ECL2 in GPR161 penetrates into the receptor pocket, and is completely occluded from the solvent, whereas the other β strand faces the solvent (Fig. 4b and Supplementary Fig. 6a). In the solvent-accessible strand, E170$^{ECL2}$, and D172$^{ECL2}$ potentially form salt bridge interactions with R184$^{ECL2}$ and K298$^{ECL3}$ respectively (Fig. 4b and Supplementary Fig. 4f); F171$^{ECL2}$ makes van der Waals contact with the backbone and W93$^{23.50}$ in ECL1. K175$^{ECL2}$ in the β-turn potentially engages two salt bridges with E293$^{6.58}$ and E306$^{7.35}$. M177$^{ECL2}$ in the other strand is inserted into a hydrophobic pocket formed by V80$^{2.57}$, V84$^{2.61}$, Y107$^{3.32}$, and F314$^{7.43}$. Consistently, mutation of Y107$^{3.32}$, F171$^{ECL2}$, or K175$^{ECL2}$ completely suppressed the basal activity of GPR161; and mutation of E170$^{ECL2}$ or E293$^{6.58}$ significantly reduced its basal activity (Fig. 4c). However, it is unclear how mutation of F174$^{ECL2}$

or R184$^{ECL2}$ can enhance the basal activity. All these mutants except M177$^{ECL2}$A are expressed at comparable levels with WT. Instead of forming β hairpin structure, ECL2 in the structure of GPR61 and previously reported structures of GPR21, GPR52, and GPR12 are organized into a short loop structure (Fig. 4d). Nevertheless, ECL2 in GPR61 shows shallower penetration in the orthosteric pocket compared to that in GPR21 and GPR52 (Supplementary Fig. 6b). Major contacts between ECL2 and the other parts of GPR61 are limited to the residues, namely S196, L197, and W199 (Fig. 4e and Supplementary Fig. 5f). S196$^{ECL2}$ make a hydrogen bond with E323$^{7.35}$; L197$^{ECL2}$ penetrates into a hydrophobic pocket formed by V123$^{3.33}$, L119$^{3.29}$, F212$^{5.42}$, and H307$^{6.55}$; and W199$^{ECL2}$ makes van der Waals contact with the backbone of TM4. Mutation of S196$^{ECL2}$, L197$^{ECL2}$, or W199$^{ECL2}$ dramatically reduced the basal activity of GPR61 (Fig. 4f), and combined mutations of residues in ECL2 (S196$^{ECL2}$G/L197$^{ECL2}$G/Q198$^{ECL2}$G/W199$^{ECL2}$G, ΔECL2) completely impaired the basal activity of GPR61. In contrast, mutation of Q198$^{ECL2}$ alone increased the basal activity. We speculate that the interaction between Q198$^{ECL2}$ and H307$^{6.55}$ limits the outward movement of TM6. Therefore, abolishing their interactions may lower the energy barrier of the active state. These mutations had minimal or no effect on receptor expression level compared with WT

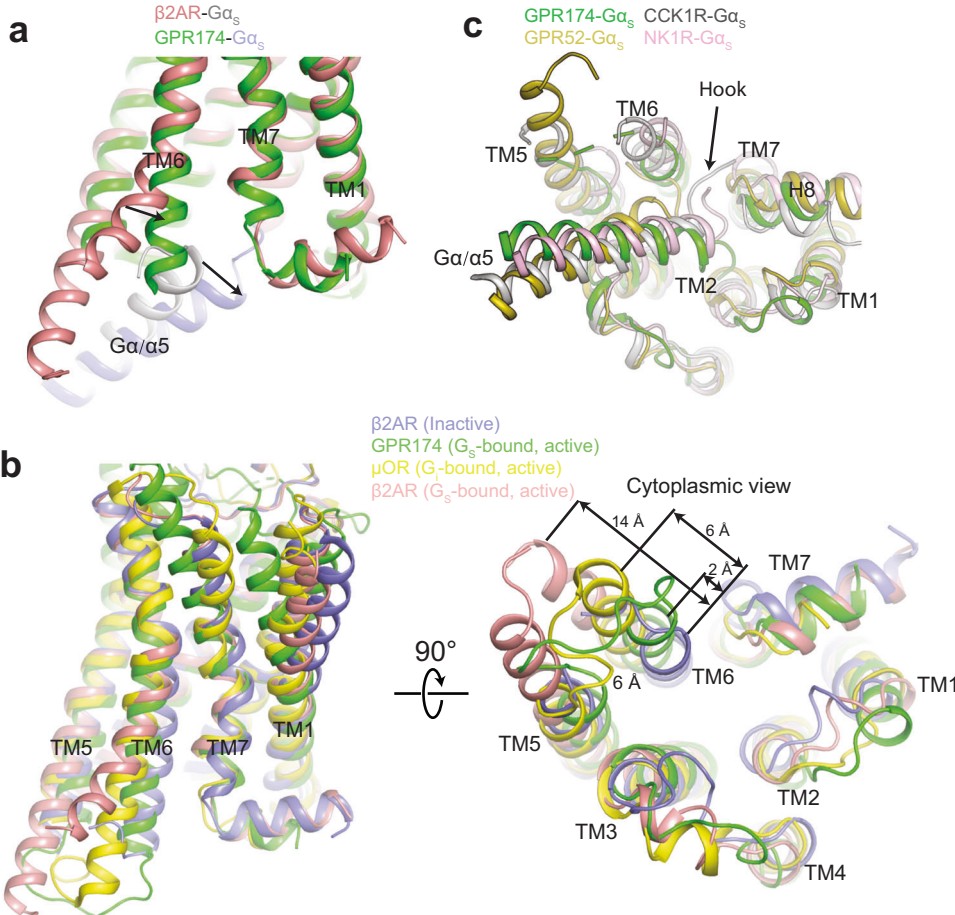

**Fig. 5 | In contrast to GPR61, GPR161, and β2AR, GPR174 adopts a non-canonical G_s coupling mode. a** Structural overlay of β2AR–G_s and GPR174–G_s complexes from the side view. **b** Comparison of structures of inactive β2AR (PDB: 2RH1), active β2AR (PDB: 3SN6), active μOR (PDB: 6DDE) and active GPR174 in two orthogonal views. **c** Structural overlay of GPR174–G_s, CCK1R–G_s (PDB: 7MBX), GPR52–G_s (PDB: 6LI3), and NK1R–G_s (PBD: 7RMI) complexes from the cytoplasmic view. All these complexes adopt a non-canonical G_s coupling mode where the hook of Gα_s is distorted and the outward movement of TM6 is less pronounced compared to β2AR. α5 of Gα_s is shown with other regions hidden.

(Fig. 4f). These results suggest that penetration of ECL2 in the orthosteric pocket of receptors contributes to the constitutive activity.

### Non-canonical G_s coupling mode

Both GPR61 and GPR161 adopt a canonical G_s coupling mode, which is observed in most class A G_s-coupled receptors such as β2 adrenergic receptors (β2AR)[36] and D1R[37] (Supplementary Fig. 6c). When bound to the receptors, the α5 helix of Gα_s undergoes rotational and translational movement, with the extreme C-terminus of α5 assuming a "hook" conformation. The "hook" is inserted into the intracellular cavity of the receptor and is in direct contact with TM6 and TM5, leading to an outward movement of TM6 and TM5. Owing to the bulkier size of the "hook" in Gα_s, the outward movement of TM6 in G_s-coupled GPCRs is more pronounced than in GPCRs that couple to the other G protein subtypes. Strikingly, when bound to GPR174 instead of contacting TM6, the "hook" is distorted and protruded into a groove formed by TM1, TM2, and TM7. Compared with inactive β2AR, TM6 in GPR174 moves outward by only 2 Å, which is smaller than the outward displacement of TM6 in the active β2AR (14 Å) and the active μ opioid receptor that is bound to Gα_i (6 Å)[38] (Fig. 5b). The less pronounced movement of TM6 in active G_s-coupled receptors can be considered as a non-canonical G_s coupling mode, which has been observed in GPR21/52[8], neurokinin 1 receptor (NK1R)[39,40] and cholecystokinin 1 receptor (CCK1R)[41,42] (Fig. 5c). Although they share a similar extent of the outward movement of TM6, they induce various conformational changes

of the "hook". Comparison of structures of GPR174–G_s and β2AR–G_s complexes reveals unique sequence features in GPR174 that determine the non-canonical coupling. Our previous studies have identified an A/V^{5.65}Φ^{5.69}Φ^{5.72} (Φ represents hydrophobic residues) motif at TM5 that is prevalent in the majority of G_s-coupled receptors including most caoGPCRs (Fig. 6a)[37]. In receptors such as β2AR and D1R that adopt canonical G_s coupling mode, A^{5.65} is projected into a hydrophobic pocket formed by L394^{(−1)}, L393^{(−2)}, and L388^{(−7)} (−1 indicates the last residue) at the extreme C-terminus of Gαs, which we called a tri-leucine pocket (Fig. 6a and b). Mutation of A^{5.65} to the large hydrophobic residue leucine causes a steric clash with the tri-leucine pocket, and thereby dramatically reduces the G_s coupling efficiency of D1R, while its mutation to the relatively small hydrophobic residue valine has little effect[37]. Therefore, as a result of the substitution of A^{5.65} with L^{5.65} in GPR174, the tri-leucine pocket observed in the canonical G_s coupling mode collapses due to steric clash with L^{5.65}, leading to a distorted "hook". Instead, L212^{5.65} in GPR174 forms strong hydrophobic interactions with L388^{(−7)} (Fig. 6b). In contrast to D1R that prefers a small hydrophobic residue (A or V) at position 5.65, mutation of L212^{5.65} in alanine remarkably reduces the G_s coupling efficiency either in the context of GPR174WT or GPR174/Y99A mutant (Fig. 6c and d). Meanwhile, GPR174 acquires unique sequence features to accommodate tri-leucine residues. D/E^{3.49} in the D/E^{3.49}R^{3.50}Y^{3.51} motif that exists in most class A GPCRs is replaced by R in GPR174. L394^{(−1)} is positioned in the hydrophobic pocket formed by the aliphatic part of R115^{3.49}, R116^{3.50}

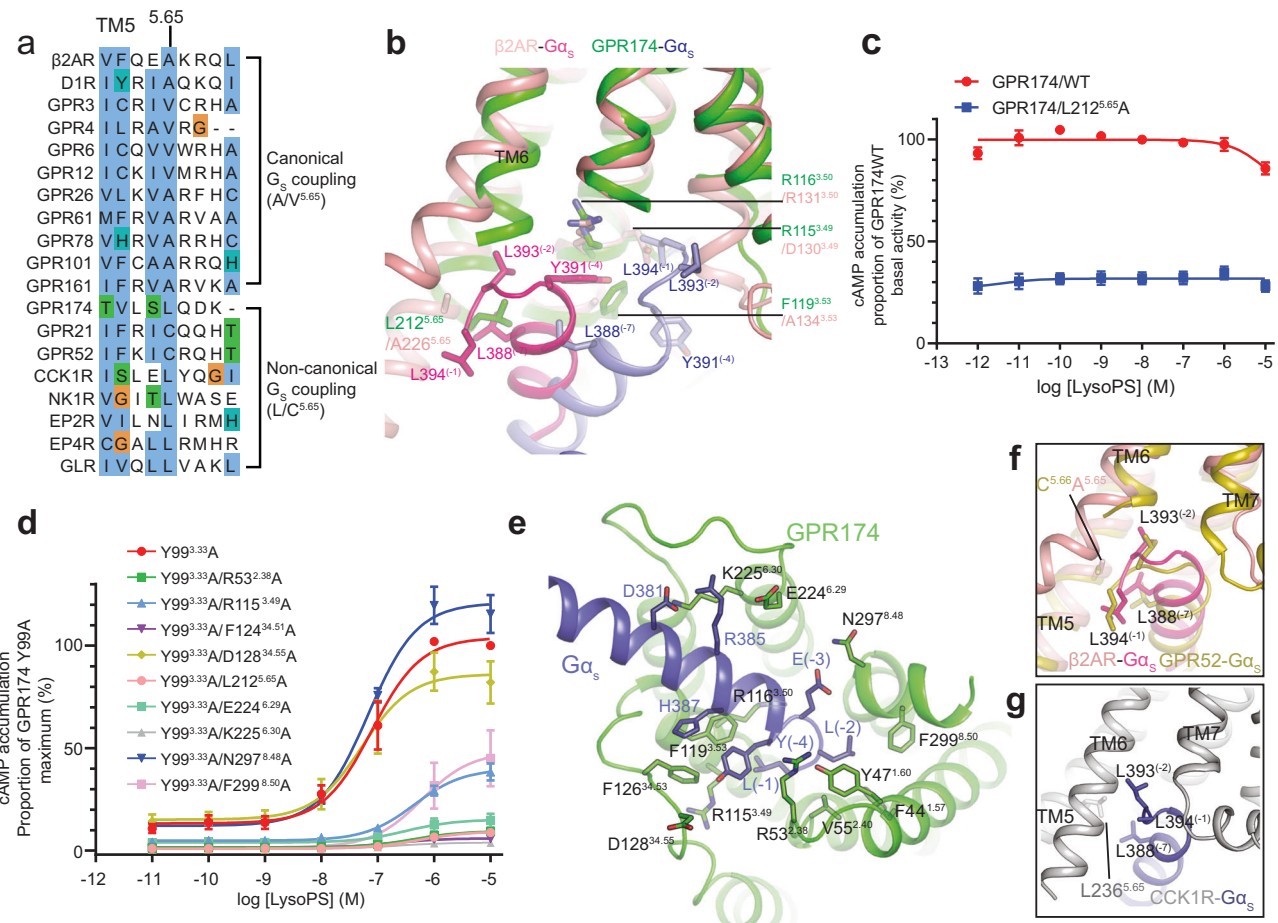

**Fig. 6 | The non-canonical G$_s$ coupling mode is determined by a larger hydrophobic residue at position 5.65 of TM5. a** Receptors that adopt canonical Gs coupling mode have a small hydrophobic residue A/V at position 5.65, while receptors that adopt non-canonical Gs coupling prefer a large hydrophobic residue (L/C$^{5.65}$). **b** Comparison of the binding interface between β2AR–Gα$_s$ and GPR174–Gα$_s$. The hook of Gα$_s$ is distorted when bound to GPR174 due to a potential steric clash between L212$^{5.65}$ and the tri-leucine pocket formed by L394$^{(-1)}$, L393$^{(-2)}$, and L388$^{(-7)}$. **c** Mutation of L212$^{5.65}$ in GPR174 remarkably reduces its basal

activity. Each data point represents mean ± SEM from three independent experiments. Source data are provided as a Source Data file. **d** The effects of mutations in the GPR174–G$_s$ interface on the potency and maximum effect of lysoPS in the context of the Y99$^{3.33}$A mutant evaluated by the cAMP assay. Each data point represents mean ± SEM from three independent experiments. Source data are provided as a Source Data file. **e** Detailed interactions between GPR174 and Gα$_s$. **f** and **g** The presence of a larger hydrophobic residue (L or C) at 5.65 in GPR52 (**f**) and CCK1R (**g**) leads to a non-canonical G$_s$ coupling mode.

and F119$^{3.53}$, while L393$^{(-2)}$ is projected into a hydrophobic groove formed by F44$^{1.57}$, Y47$^{1.60}$, V55$^{2.40}$, and F299$^{8.50}$ (Fig. 6e). Instead of contacting with R$^{3.50}$, Y391$^{(-4)}$ engages hydrophobic interactions with R115$^{3.49}$, F119$^{3.53}$, and F126$^{34.53}$ in GPR174. Moreover, D381 and R385 at α5 make additional salt bridge interactions with K225$^{6.30}$ and E224$^{6.29}$, respectively; R53$^{2.38}$ forms a hydrogen bond with the backbone of α5 (Fig. 6e). In agreement with structure observations, mutation of R53$^{2.38}$ or K225$^{6.30}$ dramatically reduced the basal activity of GPR174 (Supplementary Fig. 6g). Furthermore, we analyzed the effect of mutations on the potency and $E_{max}$ of lysoPS in the context of the GPR174/Y99A mutant which can be activated in a dose-dependent manner by lysoPS (Fig. 6d). Consistently, mutation of R53$^{2.38}$, E224$^{6.29}$ or K225$^{6.30}$ almost completely abolished G$_s$ coupling, and mutations of R115$^{3.49}$ or F299$^{8.50}$ reduced the potency by almost ten-fold and $E_{max}$ by about 50% compared with the Y99A mutant (Fig. 6d and Supplementary Table 3). Noteworthily, G$_s$-coupled receptors such as GPR21, GPR52, CCK1R, NK1R, prostaglandin receptor EP4R[43] and EP2R[44] that adopt the non-canonical G$_s$ coupling mode all possess a larger hydrophobic residue (L or C) at the position 5.65 than A/V (Fig. 6a, f, and g and Supplementary Fig. 7a–c). In the GPR52–G$_s$ complex, the tri-leucine pocket in G$_s$ moves downward to avoid steric clash with C$^{5.66}$ (equivalent to the position 5.65 in other GPCRs due to an extra residue in TM5 of GPR52),

making the cytoplasmic end of TM6 in GPR52 flexible (Fig. 6f). L394$^{(-1)}$ in CCK1R–G$_s$ and NK1R-G$_s$ are oriented outside the receptor core to avoid a steric clash with L236$^{5.65}$ (Fig. 6g and Supplementary 7a). Similar to GPR174–G$_s$, the "hook" in the EP2R–G$_s$ and EP4R–G$_s$ protrudes into a hydrophobic groove formed by TM1, TM2, and TM7 (Supplementary Fig. 7b, c). In addition, most class B G$_s$-coupled receptors including glucagon receptor (GLR) and corticotropin-releasing factor receptor also have a larger hydrophobic residue L at 5.65, which is accommodated by a more pronounced movement of TM5 and TM6, triggered by a sharp kink at the conserved PXXG motif in the middle of TM6[45–47] (Supplementary Fig. 7d). The absence of the PXXG motif in class A G$_s$-coupled receptors limits the further outward movement of TM6, thereby leading to a distorted hook when bound to class A G$_s$-coupled receptors that possess a larger hydrophobic residue at 5.65. Notably, most receptors including GPR174, GPR21, GPR52, CCK1R, and NK1R that adopt non-canonical G$_s$ coupling mode have been reported to promiscuously couple to other G protein subtypes[9,39–42]. Our previous studies have shown that a larger hydrophobic residue (L) at 5.65 is dominant in G$_q$- and G$_{i/o}$-coupled receptors[37,48]. Mutation of L214$^{5.65}$ in galanin receptor 2 to alanine almost abolished G$_q$ recruitment, indicating its critical role for G$_q$ coupling[48]. Despite differences in conformational changes of TM6 and

the hook in Gα between canonical and non-canonical $G_s$ coupling mode, a common feature of GPCRs and $G_s$ coupling is the insertion of a large hydrophobic residue at the position 34.51 of ICL2 into a hydrophobic groove in the Ras domain of Gα (Fig. 1b and Supplementary Fig. 6d–f). Mutation of $F124^{34.51}$ in GPR174 completely abolished $G_s$ coupling (Fig. 6d and Supplementary Fig. 6g). Taken together, these data suggest that the non-canonical $G_s$ coupling mode is determined by a larger hydrophobic residue (L or C) at position 5.65 of receptors, which may be critical for coupling to other G protein subtypes, whereas receptors that adopt canonical $G_s$ coupling mode prefer a small hydrophobic residue (A or V) at this position.

## Discussion

Here, we found a group of class A $G_s$-coupled oGPCRs named caoGPCRs show extremely high basal activity, compared with the other oGPCRs. The cryo-EM structure of GPR174, one of the caoGPCRs in complex with $G_s$ without exogenous ligands revealed that endogenous lysoPS had occupied the receptor, leading to maximal activation and making it not respond to exogenous lysoPS in the cAMP assay. The perfect shape and charge complementarity between lysoPS and GPR174 account for their specific recognition and high-affinity binding. Occupancy of the receptor by endogenous lysoPS is consistent with functional studies showing that lysoPS needs to be added in micromolar concentrations to promote GPR174-mediated suppression of T cell proliferation[20–23] and with the finding that exogenous lysoPS fails to increase the GPR174-$G_s$-PKA dependent up-regulation of CD86 in cultured B cells[19]. LPs including lysoPC, lysoPE, and lysoPS are synthesized via cleavage of one acyl group from membrane lipids by phospholipases. LysoPC is the most abundant LP in plasma, with a concentration reaching hundreds of micromolars, while the concentration of lysoPS is in the hundreds of nanomolar range[27]. This amount of lysoPS would be anticipated to induce full activation of GPR174 in terms of the $G_s$ pathway. However, LPs can be rapidly metabolized in tissues and interstitial concentrations are often considerably lower than in plasma[21,27,49]. In contrast to in vitro culture systems, exogenous lysoPS can induce GPR174-dependent CD86 expression in vivo[19], suggesting that lysoPS may not fully occupy GPR174 in some microenvironments. Thus, the baseline amount of GPR174 signaling in cells is likely to reflect the local balance of lysoPS synthetic and degradative activity. Increases in lysoPS generation under inflammatory conditions are likely to increase GPR174 signaling and function[21,23,50].

Liang et al. published the cryo-EM structure of GPR174–$G_s$ purified from insect cells with exogenous lysoPS[51] after we submitted this work. Our structure shows an almost identical binding pose of lysoPS in GPR174 and a similar Gs coupling mode with theirs, further supporting the identity of lysoPS in our structure, and that the GPCR-$G_s$ fusion protein strategy for structural characterization does not influence the binding mode of Gα. The weak NanoBiT $G_s$ dissociation response mentioned in Liang's work is also probably due to the maximal activation of GPR174 by the endogenous lysoPS.

While ECL2 is penetrated into the orthosteric pocket of GPR61 and GPR161 as well as GPR12, GPR21, and GPR52, and possibly contributes to their constitutive activity, we cannot rule out the possibility that the existence of endogenous ligands that may dissociate after harsh purification procedure lead to extremely high basal activity. Residues on ECL2 that show reduced basal activity when mutated may be involved in ligand binding or be required for proper folding of the receptors. As G protein coupling to GPCRs stabilizes a "closed" receptor conformation which impedes agonist binding as well as prevents the dissociation of bound ligands from the receptor[26], the penetration of ECL2 in the orthosteric binding pocket observed in these GPCRs–$G_s$ complexes could be attributed to G protein coupling to the receptors in the absence of agonists. As mentioned above, all caoGPCRs share relatively high sequence similarity with GPCRs known to be activated

by lipids. ECL2 in EP2R occupies a similar position as that in GPR161, with the endogenous ligand PGE2 bound in the side pocket[44] (Supplementary Fig. 6a), suggesting that unknown endogenous lipids may bind GPR161 in a similar manner. Based on their sequence homology with S1PR, GPR3/6/12 accommodate a conserved hydrophobic pocket for binding with the acyl chain of lipids (Supplementary Fig. 7e). Previous studies have shown that S1P or SPC can induce intracellular $Ca^{2+}$ mobilization in cells expressing GPR3, GPR6, or GPR12[14]. However, controversial pharmacological data were obtained using different functional assays such as the cAMP assay or β-arrestin recruitment assay with different cell lines. In the case of GPR174, owing to the higher potency of lysoPS in the $G_s$ pathway than $G_{12/13}$, the exogenous lysoPS cannot further increase the cAMP level in our cell line but can activate GPR174 in a dose-dependent manner using the TGFα shedding assay. GPR174 mutants with a reduced potency of lysoPS can respond to exogenous lysoPS in the cAMP assay. Therefore, different potencies among distinct signaling assays and different concentrations of ligands released by different cell lines or present in different cell media can cause functional discrepancies. However, it is very challenging to identify the endogenous ligands of caoGPCRs, since most of them may have been fully activated by the endogenous ligands in the $G_s$ pathway and no longer respond to the exogenous ones. Structural approaches in complement with functional assays will be important to identify and confirm what the endogenous ligands for caoGPCRs are.

Unlike conventional GPCRs in which their activity is spatially and temporally controlled by the release of ligands such as neurotransmitters under certain stimulation, the caoGPCRs are partially or fully activated once expressed in vivo. The question is how the activity of caoGPCRs is precisely modulated in vivo. In adipose tissue, GPR3 is expressed at a very low level, and cold exposure triggers transcription of *Gpr3* that drives thermogenesis via $G_s$[52]. GPR161 is trafficked to primary cilia through the Tulp3/IFT-A complex[29], which is essential for inhibiting Shh signaling and normal embryonic development. The transcriptional induction of GPR3 and trafficking of GPR161 to cilia are analogous to ligand-regulated activation of conventional GPCRs. The activity of caoGPCRs can be terminated via receptor internalization and degradation. Moreover, the existence of endogenous antagonists (agouti family) for melanocortin receptors[53] raises the possibility that the endogenous antagonists may exist in vivo and modulate the activity of caoGPCRs. To investigate how the activity of caoGPCRs is modulated will be an exciting research area in the future.

## Methods
### Cloning

All GPCR sequences used in this study were subcloned from the PRESTO-Tango GPCR Kit[54] into the pcDNA3.1 vector, with an N-terminal hemagglutinin (HA) signaling peptide (MKTIIAL-SYIFCLVFA) following by a FLAG tag (DYKDDDDA). For the NanoBiT mini-$Gα_s$ recruitment assay, the smBiT[55] tag (VTGYRLFEEIL) was added to the C-terminus of GPCRs with a flexible linker (GGGSGGGGSGGSSSGG) in between, and the lgBiT tag was fused to the N-terminus of mini-$Gα_s$[56]. For GPR61–, GPR161–, or GPR174–mini-$Gα_s$ fusion proteins, the C-terminus of GPCRs was fused with the N-terminus of mini-$Gα_s$399 with a 34 amino acid linker. All point mutations were introduced by quick change following standard procedure.

### Expression and purification of Gβ1γ2 and Nb35

The human His$_6$-tagged Gβ$_1$ and Gγ$_2$ with a C68S mutation were cloned into pFastBac™ Dual vector and expressed in Sf9 incest cell using Bac-to-Bac Baculovirus expression system (ThermoFisher Scientific). For protein expression, the Sf9 cell was infected at $3 \times 10^6$ cell/ml density. After 48 h, cells were harvested by centrifugation, and the pellet was resuspended in wash buffer I (20 mM HEPES, pH 7.4, 150 mM NaCl, and 20 mM imidazole) supplemented with 5 mM CaCl$_2$ and 0.5 mM NiSO$_4$,

and were by Dounce homogenizer and sonication. The homogenate was centrifuged at 35,000×*g* at 4 °C for 30 min. The supernatant was filtered through a 0.45 μm membrane filter, and loaded onto Ni-NTA resin by gravity. Ni beads were washed by wash buffer, and $G\beta_1\gamma_2$(C68S) protein was eluted by elution buffer (20 mM HEPES, pH 7.4, 150 mM NaCl, 250 mM imidazole), and further purified by HiTrap Q HP anion exchange chromatography column.

For Nb35 protein purification[36], a plasmid encoding Nb35 with a periplasmic signal peptide was transformed into *Escherichia coli* strain BL21 (DE3). 2 L bacteria were cultured in LB media until $OD_{600}$ reached 0.6–0.8. Bacteria were further shaken at 37 °C for 12 h after the addition of 500 mM isopropyl 1-thio-β-D-glucopyranosided (IPTG). Bacteria were collected by centrifugation, resuspended in 100 ml SET buffer (0.5 M Sucrose, 0.5 mM EDTA, 0.2 M Tris, pH 8.0), and stirred at room temperature (RT) for 45 min. The lysate was added with 200 ml $H_2O$ to induce osmotic shock and stirred for 45 min before adding a final concentration of 150 mM NaCl, 2 mM $MgCl_2$, and 20 mM Imidazole. The lysate was clarified by centrifugation and loaded onto Ni-NTA resin followed by the same purification procedure as above. After elution from nickel resin, Nb35 was further purified by a Superdex 75 increase 10/300 GL column. Both $G\beta_1\gamma_2$(C68S) and Nb35 were concentrated to 1 mg/ml, flash frozen in liquid nitrogen, and stored at −80 °C.

## Expression and purification of GPR61/GPR161/GPR174–mini-Gαs fusion protein

All GPCRs were transiently expressed in Expi293F cells using a serum-free SMM 293-TII expression medium (SinoBiological). 500 ml cells were transfected with 500 μg plasmids using polyethyleneimine "MAX" (Polysciences, Inc) at $2.3 \times 10^6$ cells per ml density. Cells were harvested by centrifugation at 48 h post-transfection, and lysed in in hypotonic buffer containing 25 mM HEPES (pH 7.4) and 50 mM NaCl by glass dounce. Membranes were pelleted by centrifugation at 38,000×*g* and solubilized in solubilization buffer (20 mM HEPES pH 7.4, 150 mM NaCl, 2 mM $CaCl_2$, 0.5% LMNG, and 0.05% CHS). Homogenate was stirred at 4 °C for 2 h before centrifugation at 35,000×*g* for 30 min at 4 °C to remove debris. Supernatant was filtered through a 0.45 μm membrane filter, and loaded onto M1-Flag agarose beads. The beads were washed with wash buffer II (20 mM HEPES pH 7.4, 150 mM NaCl, 2 mM $CaCl_2$, 0.01% LMNG, 0.001% CHS). The bound receptors were eluted with elution buffer (20 mM HEPES pH 7.4, 150 mM NaCl, 5 mM EDTA, 0.1 mg/ml Flag peptide, 0.01% LMNG, and 0.001% CHS). The eluted proteins were concentrated and loaded onto a Superose 6 increase 10/300 GL column to further remove aggregation.

## Complex assembly

For complex assembly, the purified GPCRs–mini-$Gα_s$ fusion protein, $G\beta_1\gamma_2$, and Nb35 were mixed at a molar ratio of 1:1.2:1.2 supplemented with 2 mM $MgCl_2$, 1 μl apyrase, and 1 μl 10% LMNG stock solution and incubated on ice overnight. The complexes were purified by a Superose 6 increase 10/300 GL column in SEC buffer (20 mM HEPES pH 7.4, 150 mM NaCl, 0.01% LMNG, and 0.001% CHS). The peak fractions corresponding to the complex were pooled and concentrated to 7–8 mg/ml for cryo-EM analysis.

## Cryo-EM sample preparation and data collection

3.0 μl of protein complex was loaded onto a glow-charged Quantifoil Au R1.2/1.3 300 mesh grids and incubated for 8 s in a chamber maintained at 8 °C and 100% humidity, blotted for 3.0–5.0 s and plunge frozen into liquid ethane using a FEI Vitrobot MarkIV (Thermo Fisher Scientific). Cryo-EM movies for all complexes were collected on a Titan Krios (Thermo Fisher Scientific) at a nominal magnification of ×64,000 equipped with a BioQuantum GIF/K3 (Gatan) direct electron detector in a superresolution mode. Data were collected using EPU at a dose rate of 22 e⁻ pixel⁻¹ s⁻¹ for a total dose of 50 e⁻ Å⁻² over 32 frames. Cryo-

EM data collection parameters are summarized in Supplementary Table 2.

## Cryo-EM data processing

A total of 923 movies were collected for the GPR61–miniGα_s–$G\beta_1\gamma_2$–Nb35 complex, 1058 movies for the GPR161–miniGα_s–$G\beta_1\gamma_2$–Nb35 complex, and 2043 movies for the GPR174–miniGα_s–$G\beta_1\gamma_2$–Nb35 complex. Movies were motion-corrected and 2× binned to a pixel size of 1.08 Å using MotionCor2[57]. Contrast transfer function (CTF) was estimated using patch-based CTF estimation in cryoSPARC[58]. Then the particles were picked by a Blob picker with a 180-pixel box size and subjected to a round of 2D classification in cryoSPARC. 2D class averages with clear secondary structural features were combined and used as a template for another round of particle picking by a Template picker. The selected particles were subjected to several rounds of 2D classification, a round of Ab initio reconstruction, and a round of heterogenous refinement. Finally, non-uniform (NU) refinement and local refinement were performed in cryoSPARC to obtain density maps with global resolution of 3.16 Å for the GPR61–mini-Gα_s–$G\beta_1\gamma_2$–Nb35 complex, 3.10 Å for the GPR161–miniGα_s–$G\beta_1\gamma_2$–Nb35 complex and 2.83 Å for the GPR174–mini-Gα_s–$G\beta_1\gamma_2$–Nb35 complex.

## Model building

Structural models of GPR61, GPR161, and GPR174 were generated by AlphaFold[59]. The structures of the receptor and the mini-Gα_s–$G\beta_1\gamma_2$–Nb35 complex (PDB: 7F0T) are fitted into the density map using UCSF Chimera[60,61] to generate the atomic model for the full complex. All three models were manually adjusted in COOT 0.9-pre and refined in Phenix[62] using the secondary structure restraints. Structure and density map figures were prepared by Pymol and UCSF Chimera X[60].

## NanoBiT mini-Gαs recruitment assay

NanoBiT[55] assay is a NanoLuc-based enzyme complementation system, in which the large fragment (LgBiT) element is fused to the N-terminus of mini-$Gα_s$ proteins, and the small fragment (SmBiT) is fused to the C-terminus of GPCRs. Expi293F cells cultured in the serum-free SMM 293-TII medium were seeded into a six-well plate at a density of $2 \times 10^6$ cells per ml and transfected with 1 μg of GPCR-smBit plasmid and 1 μg of mini-$Gα_s$-lgBit plasmid using PEI. After 24 h, cells were centrifuged, washed once with Hank's balanced salt solution (HBSS), and resuspended into the NanoBiT assay buffer (HBSS supplemented with 0.01% bovine serum albumin (BSA, Sigma), 10 mM HEPES (pH 7.3, Beyotime) and 10 μM coelenterazine-h (Yeasen)). Cells were seeded into a black-bottom 96-well plate and equilibrated at room temperature (RT) for 1 h. Relative bioluminescence units (RLUs) were measured by Spark multimode microplate reader (Tecan).

## cAMP accumulation assay

In the cAMP accumulation assay, Expi293F cells cultured in the serum-free SMM 293-TII medium were seeded into a six-well plate at a density of $2 \times 10^6$ cells per ml and transfected with 1 μg of GPCR plasmid and 1 μg of pGloSensor-22F cAMP plasmid (Promega). At 24 h post-transfection, cells were washed once with HBSS and resuspended into the cAMP assay buffer (HBSS supplemented with 0.01% bovine serum albumin (BSA, Sigma), 10 mM HEPES (pH 7.3, Beyotime) and 500 μg/ml D-luciferin (Beyotime)), and then seeded into black-bottom 96-well plate with 99 μl cells. After incubation at RT for 30 min, cells were stimulated using 1 μl of ligands with titrated concentrations, and RLUs were measured by Spark microplate reader in 3–5 min. Data were fitted to a three-parameter sigmoidal concentration–response in GraphPad Prism 7.0.

## NanoBit Gs recruitment assay

2 ml Expi293F cell was seeded into a six-well plate at a density of $2 \times 10^6$ cells/ml and transfected with five plasmids (200 ng GPCR-SmBit, 200 ng WT-Gαs-LgBit, 500 ng Gβ1, 500 ng Gγ2 and 100 ng RIC8A) using PEI. LgBit is inserted between L113 and V114 of Gαs, with a 15 aa linker in between. CMV promoter in pcDNA3.1 driving GPCR expression was changed to a weaker expression promoter spleen focus-forming virus (SFFV). 24 h after transfection, the cell was centrifuged at $100 \times g$ for 3 min and washed by HBSS 2 times. Then, the cell pellet was resuspended into 4 ml NanoBiT assay buffer and seeded into a white bottom 96-well plate with 95 μl cells. After incubation for 30–60 min, 5 μl of ligand was added into each well, and RLUs were measured by Spark microplate reader in 3–5 min.

## Flow cytometry

2 ml of Expi293F cells were transfected by Flag-tagged GPCR plasmids by PEI at $2.3 \times 10^6$ cells per ml density and cultured in a six-well plate. After 24 h, 100 μl of cells were centrifuged at $250 \times g$ for 3 min to remove the culture medium and washed twice with the FACS buffer (20 mM HEPES, 150 mM NaCl, 2 mM $CaCl_2$, 0.01% BSA, pH 7.4). Cells were resuspended in FASC buffer supplemented with 1 μl of Alexa Fluor 647-conjugated M1-Flag antibody for 15 min and washed once by the FACS buffer to remove free labeled antibody before analysis using flow cytometry (BD Accuri™ C6 Plus). APC-derived fluorescent signal was recorded in an FL4 channel of the flow cytometry and analyzed by FlowJo 10 software.

## MS analysis

240 μl of methanol was added to 60 μl of 29 μM GPR174–mini-Gαs–Gβ₁γ₂–Nb35 or D1R–mini-Gαs–Gβ₁γ₂–Nb35, mixed by vortex and centrifuged at $14{,}000 \times g$ at 4 °C for 15 min. The supernatant was transferred to liquid chromatography (LC) vials with a micro-insert and 3 μl was injected into LC–MS. LC–MS analysis was performed on a Thermo Vanquish UHPLC equipped with a Thermo Q Exactive HF-X hybrid quadrupole-Orbitrap mass spectrometer in negative ESI mode. A Waters Acquity CSH C18 column (2.1 × 100 mm, 1.7 μm) was used for separation. The mobile phase contains 60:40 acetonitrile:water (A) and 90:10 isopropanol:acetonitrile (B), both with 10 mM ammonium acetate. The following gradient was applied: 0–2 min,15–30% B; 2–2.5 min, 30–48% B; 2.5–11 min, 48–82% B; 11–11.5 min, 82–99% B; 11.5–12 min, 99% B; 12–12.1 min, 99–15% B; 12.1–15 min, 15% B. Column temperature was maintained at 65 °C and the flow rate was 0.6 ml/min. Full-scan mass spectra were acquired in the range of $m/z$ 120–1800 with the following ESI source settings: spray voltage: 2.5 kV, auxiliary gas heater temperature: 380 °C, capillary temperature: 320 °C, sheath gas flow rate: 30 units, auxiliary gas flow: 10 units. MS1 scan parameters included resolution 60000, AGC target 3e6, and maximum injection time of 200 ms. MS/MS data was collected with a normalized collision energy (NCE) of 20. Data processing was performed with MS-DIAL[63] software (v. 4.16).

## MD simulations

Six simulation systems were prepared using the Membrane Builder module in CHARMM-GUI with cryo-EM structure as an initial model. GPR174 was inserted in an explicit bilayer of 1-palmitoyl-2-oleoyl-sn-glycerol-3-phosphocholine (POPC) lipids. A 22.5 Å water layer was added to the top and bottom of the lipid bilayer. 150 mM of $Na^+$ and $Cl^-$ ions were added to the bulk water to achieve a charge-neutral condition in the restrained equilibration. In all the simulations, the systems were parameterized using the CHARMM36 force field[64] for protein, lipids, and water; and CGenFF[65] for the small molecules. All simulations were performed using the GROMACS (2021.4)[66]. The systems were minimized for 5000 steps with the steepest descent algorithm. Six turns of 500 ps restrained equilibration were carried out to relax the system following the CHARMM-GUI protocol[67]. After equilibration, a 100 ns NPT production run was performed for each system. All simulations were performed under the following protocol. A 2-fs time-step with the LINCS algorithm was used. The van der Waals interactions were smoothly switched off at 10–12 Å by a force-switching function, and long-range electrostatic interactions were calculated using the particle-mesh Ewald method. Temperature and pressure were held at 298.15 K and 1 bar, respectively. Temperature and pressure controls were achieved with a Nose–Hoover thermostat and Parrinello–Rahman barostat for dynamics.

## Reporting summary

Further information on research design is available in the Nature Portfolio Reporting Summary linked to this article.

## Data availability

The atomic structures of GPR174–Gs, GPR161–Gs, and GPR61–Gs have been deposited at the Protein Data Bank (PDB) under the accession codes 8KH5, 8KH4, and 8KGK. The EM maps have been deposited at the Electron Microscopy Data Bank (EMDB) under the accession numbers EMD-37237, EMD-37236, and EMD-37224. Source data are provided with this paper. Uncropped scans of all blots and gels used in Supplementary Information are shown in Supplementary Fig. 8. Source data are provided with this paper.

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

## Acknowledgements

We thank the staff at Shuimu BioSciences for their help with cryo-EM data collection. All EM images were collected at Shuimu BioSciences. This work has been supported by the Beijing Municipal Science & Technology Commission (Z201100005320012, N.H.)

## Author contributions

Y.N. and Z.Q purified the protein complexes, collected cryo-EM data and performed cryo-EM data processing and model building, and performed functional assays with the supervision of S.Z. S.C. performed NanoBiT Gs recruitment assay. X.S. and Y.M. performed MS experiments. Z.C. and N.H. performed MD simulations. S.Z. wrote the manuscripts with helpful inputs by J.G.C.

## Competing interests

The authors declare no competing interests.
