## [Peer Review File · Nature Communications]

Specific binding of GPR174 by endogenous lysophosphatidylserine leads to high constitutive Gs signalingREVIEWER COMMENTS

Reviewer #1 (Remarks to the Author):

This manuscript by Nie et al. focused on a group of orphan class A/rhodopsin-like GPCRs, including GPR61, GPR161, and GPR174, with high constitutive activity. Using single-particle cryo-EM, the authors determined the structures of GPR61-Gs, GPR161-Gs, and GPR174-Gs complexes without exogenous ligands. The structures of GPR161 and GPR61 reveal that the second extracellular loop (ECL2) penetrates into the orthosteric pocket, and the GPR174 adopts a non-canonical Gs coupling mode. In addition, the authors claimed that the GPR174 is an orphan receptor, and LysoPS is its endogenous ligand. Overall, I think this manuscript has several serious flaws to be addressed before publication. I therefore recommend for a major revision.

Major concerns:

1. The authors determined the structure of GPR174 and modeled LysoPS with 15 carbons in the acyl into the unassigned density. In my opinion, Figure 2C is misleading, since it does not match the lipidomics results. The lipidomics indicated that the isoforms from 16:0 to 20:1 are determined from Mass Spectrum analysis. The density of LysoPS in the current structure is unconvincing, the author should further confirm the binding pose of LysoPS in GPR174 by determining the structure of GPR174 with specific LysoPS molecule (LysoPS 18:1) or perform computational simulations.

2. In this manuscript, the authors would like to understand the activation mechanism of orphan receptors. However, LysoPS was already identified as an endogenous ligand for GPR174 in previous studies, this receptor doesn't belong to an orphan receptor. Please cite several important and already published articles, which demonstrate that LysoPS is the endogenous ligand of GPR174 (PMID: 22983457). In addition, the author classified these 12 caoGPCRs into four types, but without any specific criteria. It is not reasonable according to their sequence similarity as described by the author. The phylogenetic tree analysis may be one of the ways to reflect the sequence conservation. The authors can provide more information about the mechanism of caoGPCRs with high basal activity if they can determine the structures of GPR26 or GPR6. Moreover, could the author get any common features (or motifs) of caoGPCRs from the current structure or data analysis?

3. The author provided more information on GPR174 activation and identified the ligand LysoPS in the receptor. I suggest the author confirm the binding mode by other experiments (BRET assay or G protein dissociate assay).

4. The author just tested the Gs signaling for these receptors. The author should measure the constitutive activities of other signaling pathways (Gi, Gq, G12/13, arrestin). Only determination of Gs activity for these receptors is an important flaw, because GPCRs were known to couple to 20 different G protein subtypes.

5. Fig 1a shows the comparison of the signal intensity of D1R self-activation with different orphan receptors of class A receptor. This data is questionable. The receptors listed in supplementary table 1 showed varied cell surface expression levels, which will certainly lead to different downstream signaling pathway and different activities. The authors should ensure all receptors tested are expressed at similar cell surface expression levels by adjusting the plasmids amount transfected into the cells.

6. While the authors have provided PDB files for review, many figures and descriptions in the manuscript are not consistent with the provided PDB files. For example: (i) Figure 3c and lines 107-110 describe the phosphate group in lysoPS as covered by three spatially separated basic residues, the model of residues (R181.31, R1564.64, K2576.62) in GPR174-Gs don't fit with density. (ii) the authors describe many salt bridge interactions in Figure 4b and lines 150-153 between glutamic acid or aspartic acid with Arginine or Lysine, but at least two residues (K298ECL3, E3067.35) lack the density of this part, the modeling is very ambiguous with the map density in this region; (iii) In GPR174-Gs structure, there is almost no EM density around C11 and C16 of lysoPS. Figure 1c didn't reflect the fact and is misleading; (iv) In GPR61-Gs structure, the van der Waals shown in Figure 4f between the W199ECL2 and TM4, is also not supported by the model provided, with almost no density for W199 ECL2 of GPR61. Moreover, there is a numbering of modeling errors in the PDB files provided for review (poor rotamers/poor fit to the experimental density maps) that need to be addressed. I would recommend the authors thoroughly check through the PDB files with the density maps. There are also many regions (outside of the regions discussed specifically in the manuscript) within the PDB files that are modeled differently in the 3 structures, which are not supported by differences in density in the cryo-EM maps. These are generally regions that have poor density, so modeling accurately is difficult, therefore I would recommend the authors either remove these regions from the models or if modeled, model these regions the same for all structures, unless there is sufficient density to identify clear differences between the different structures. These are key issues because the data don't support their scientific claims.

7. In this article, the authors used the nanobit system to recombine the complex of GPR61-Gs, GPR161-Gs and GPR174-Gs, regardless of whether there are any physiological or functional supports for corresponding signaling complexes. The nanobit system is easy to artificially tether the complex even if they don't exist in physiological conditions. So, does this really exist under physiological conditions?

Minor concerns:

1. The cAMP accumulation effect of R156A(4.64) was reduced to ~35%, and there was no significant increase under the treatment of a higher concentration of LysoPS, which is contradicted by what is described in the text.

2. As lysine has only one amino group, I am curious whether it can form two ionic bonds with E293 and E306 at the same time. (Line152)

3. As is described in line157, "ECL2 in the structure of GPR61 and previously reported structures of GPR21, GPR52 and GPR12 are organized into a short loop structure (Fig. 4d)." The short loop structure of GPR21, GPR52 and GPR12 should also be shown, to present the similarities and differences with GPR61.

4. Q198 appears to interact with H307, however, the substitution of Q198 instead enhances the constitutive activity. It is necessary to include more discussion and come up with some reasonable explanations.

5. Control curves should be included (e.g., Fig3e and Fig6d). For example, Fig6d should include the curve of the Y99A single mutant.

6. In Figure 3c, the assumed LysoPS form extensive contacts with GPR174, but the authors only examined the contribution of R75, Y99, R156, and K257 to basal activity and signals, the other mutants are needed to test to illustrate a more detailed mechanism. Moreover, the authors must provide ELISA assay data or Western Blot data for mutants to ensure similar cell surface expression levels of WT and its mutants.

7. The GPR174-LysoPS-Gs structure has already been solved by He et al, Nat Commun, 2023, and the resolution is higher than that in this manuscript. But the authors did not include any discussions about this. Whether the structure models are similar or different? Is there any new insight?

8. The previous study (Journal of Neuroscience Research, 2009, Doi: 10.1002/JNR.21955) mentioned that the N-terminal has an important role in GPR61 constitutive activity. In the study, why only considered the role of ECL2 in its constitutive activity? The role of the N-terminal was not mentioned.

9. Line 83, "We speculated the endogenous ligands might have occupied the receptor, leading to its maximal activation in the cAMP assay". According to the results shown in fig1c, the authors have the above conjecture that the occupation of pocket by endogenous ligand will preclude the activation of the

receptor by additional ligand LysoPS. But how do the authors explain the decline in the back of the curve?

10. The density of ligands shown in Fig3b still has unassigned areas. Could the authors continue to work on the existing data to enhance the precision of ligands or consider other endogenous substances inherent in 293 cells?

11. Line 114 mentioned that, as shown in Fig3e, GPR174 had a dose-dependent signal activation on LysoPS after the four amino acids (R752.60, Y993.33A, R1564.64A or K2576.62) were mutated into alanine. But why the signal data of WT is not shown in Fig3e at the same time as the control. Comparison could not be derived from two independent experiments.

12. Fig4c and 4f clearly reflect the same type of experimental data, so it is recommended to use a combination of similar order when arranging the figure. For example, the position of "Empty" should be in the same position, to increase the beauty and rationality of the figure.

13. Does the C-terminal mini-G α s399 fusion protein affect the conformation of C-terminus of GPR174, GPR161 or GPR61? The authors should clarify by measuring their activities.

14. The order of the histograms of fig4c and 4f should be consistent, including the order of empty, WT and mutations.

15. The author used a large number of highly saturated colors for the diagrams involving structural comparisons, which could not highlight the meaning he wanted to express.

16. Whether the GPR174-mini-G α s fusion protein affect the conformation of C-terminus in mini-G α s? The authors should perform molecular dynamic simulation to confirm the distorted "hook" conformation.

Reviewer #2 (Remarks to the Author):

This study determined 3 active structures of GPR61, GPR161 and GPR174, which are orphan GPCRs with high basal activity. In addition, they also identified the endogenous ligand of GPR174 and investigated the self-activation mechanism of GPR61 and GPR161. Most importantly, they revealed a non-canonical Gs coupling mode. It is very interesting and will provide useful information for the structural studies of orphan GPCRs. I have some concerns as below before the manuscript will be considered by publication in Nature Communications.

The major ones:

- 1) For GPR174, only 4 mutants were tested but there are a lot of interactions between GPR174 and lysoPS. Did you test other mutants, for example, the residues in the deeper hydrophobic pocket?
 - 2) The part of self-activation of GPR61 and GPR161 is weak. There are some other GPCRs which are activated by ECL2, it is better that add sequence and structure comparisons of GPR61, GPR161, GPR52, GPR21 and so on.
- Besides, all the ECL2 mutations of GPR61 didn't kill the basal activity, which indicate that ECL2 may not be the only activation factor. What if replace the whole ECL2 with GSSGSS linker or deletion of the ECL2?
- 3) In the Gs binding mode part, it was mentioned that L5.65 is the key residue for the non-canonical Gs coupling. So, what is the result of L5.65 mutation? Looks like there is no experimental results for this mutation. The fig 6b is just a structural comparison and it doesn't match the statement in line 191.

Small concerns:

- 1) figure 3b, the grey density in the pocket is obviously larger than the lysoPS and it is different from fig 2c. Is it also including part of the receptor side chain density or is there any extra density in the pocket except lysoPS?
- 2) figure 3c, I noticed that Y79 is labelled but didn't mentioned in the paper, what kind of interaction of this residue?
- 3) line 124, the current data in the paper didn't test G12/G13, so maybe remove this statement.
- 4) line 178, what is the outward movement distance of TM6 in active beta2AR and μ opioid receptor? It should be a clear statement including the exact number.

Reviewer #3 (Remarks to the Author):

Nie Y. et al reported cryo-EM structures of GPR61-Gs, GPR161-Gs, and GPR174-Gs signalling complexes without exogenous ligands. By analyzing these structures, they discussed the structural features of constitutively active orphan GPCRs (caoGPCRs), and found out that these receptors can be fully activated by endogenous ligands (e.g., GPR174 showing high affinity to endogenous LysoPS), or by occupation of ECL2 in the orthosteric site in the absence of endogenous ligand (e.g., GPR161 and GPR61). The coupling mode of Gs protein was also discussed, and their results suggested a possibility that some of GPCRs (such as GPR174) are capable of coupling with other G protein subtypes rather than Gs, concluding that a non-canonical Gs coupling mode was taken. The topic is intriguing; however the contents are scientifically ambiguous.

Major Concerns:

- 1) About the definition of “constitutive active”: usually “constitutive active” means Constitutive (basal) activity is defined as “ligand independent activity, resulting in the production of a second messenger in the absence of an agonist” (Scientific Reports volume 6, Article number: 38564 (2016)). However, the title and the main topic of this paper described “ligand-dependent” constitutive activation of GPR174. This is a new concept if the observation is correct. In order to avoid the confusion, the authors may use another terminology.
- 2) The authors mentioned endogenous lysophosphatidylserine (lysoPS) always binds to GPR174 and GPR174 was activated. The authors proposed two factors: abundance of lysoPS, and high affinity of lysoPS to GPR174. So the authors will show experimentally binding constant of lysoPS to GPR174.
- 3) The authors try to exclude lipidic materials including lysoPS by addition of serum albumin and do the cAMP assay.
- 4) The authors will examine other cell lines (at least two, additionally) to check the reproducibility of the phenomenon. If there is not reproducible, the present cell line will represent a pathogenic constitutive active state of GPR174.
- 5) In the case of GPR161, the binding between ECL2 and the orthosteric pocket was discussed by mutations of residues on ECL2. The mutations of the counterpart (e.g., R184ECL2 that binds to E170ECL2, E2936.58 and E3067.35 that are interacting with K175ECL2, and the hydrophobic residues interacting with M177ECL2) should also be carried out to confirm their hypothesis.
- 6) Which isoform of LysoPS (acyl chain length) was used in the author’s cAMP accumulation assay? Please specify it.
- 7) Line 226. “Mutation of F12434.51 ...” However, the interaction between F124 in GPR174 and G α was not specified in Figure 6c. Please specify.
- 8) This year, Liang J. et al also reported the cryo-EM structure of GPR174 (Nature Communications volume 14, Article number: 1012 (2023)), which showed similar ligand binding mode and Gs binding mode and GPR174. So What is the new finding in this paper?

9) In Liang's result (Nature Communications volume 14, Article number: 1012 (2023)), Y993.33 interacts with carboxylate of serine head instead of sn-2/sn-3 hydroxy group (sn-2 hydroxy group interacts with Y246 instead). Another medicinal chemistry study by Ikubo M. et al (J. Med. Chem., 2015, 58 (10), pp 4204-4219) suggested the critical role of sn-2 hydroxy group in GPR174 activity. How can we explain the agonistic activity of Y99A GPR174 mutant induced with respect to exogenous LysoPS?

10) The authors need to more discuss the relationship between the non-canonical binding of Gs protein and constitutive active state.

In summary, the contents need to be checked scientifically rigorously. The present version of the manuscript is speculative and lacks the rigid conclusion.

Reviewer #1 (Remarks to the Author):

This manuscript by Nie et al. focused on a group of orphan class A/rhodopsin-like GPCRs, including GPR61, GPR161, and GPR174, with high constitutive activity. Using single-particle cryo-EM, the authors determined the structures of GPR61-Gs, GPR161-Gs, and GPR174-Gs complexes without exogenous ligands. The structures of GPR161 and GPR61 reveal that the second extracellular loop (ECL2) penetrates into the orthosteric pocket, and the GPR174 adopts a non-canonical Gs coupling mode. In addition, the authors claimed that the GPR174 is an orphan receptor, and LysoPS is its endogenous ligand. Overall, I think this manuscript has several serious flaws to be addressed before publication. I therefore recommend for a major revision.

Major concerns:

1. The authors determined the structure of GPR174 and modeled LysoPS with 15 carbons in the acyl into the unassigned density. In my opinion, Figure 2C is misleading, since it does not match the lipidomics results. The lipidomics indicated that the isoforms from 16:0 to 20:1 are determined from Mass Spectrum analysis. The density of LysoPS in the current structure is unconvincing, the author should further confirm the binding pose of LysoPS in GPR174 by determining the structure of GPR174 with specific LysoPS molecule (LysoPS 18:1) or perform computational simulations.

Response: Thanks for bringing this up! It is not surprising we can only observe EM density for lysoPS with 15 carbons even though 16:0 to 20:1 are highly enriched. We think that the carbons that are absent in our structure are likely to be very flexible due to the lack of contact with the receptor. We have added discussion about this in the paper.

In the recent published structure of GPR174-Gs complex, although Liang et al., have already used lysoPS with 18 carbons for EM studies, the density for the first four carbons is not continuous.

2. In this manuscript, the authors would like to understand the activation mechanism of orphan receptors. However, LysoPS was already identified as an endogenous ligand for GPR174 in previous studies, this receptor doesn't belong to an orphan receptor.

Response: Although endogenous ligands have been shown to activate some orphan GPCRs, they are still classified as orphan in GPCRdb. One reason is that there is lack of sufficient evidence. For GPR174, in addition to lysoPS, other group has reported CCL21 is a potential ligand (PMID: 31875850). Dr. Jason Cyster group found that high concentrations of lysoPS are required to promote GPR174-mediated suppression of T cell proliferation, arguing whether other endogenous ligands of GPR174 with high potency exist. Moreover, our work showed that lysoPS fails to stimulate cAMP accumulation in cells expressing GPR174 using the Glosensor cAMP assay. All these issues can be addressed in our study. We found that lysoPS among all endogenous lipids is specifically copurified with GPR174, supporting that lysoPS is the most potent ligand among all lipids for GPR174. The structure further shows perfect shape complementarity and extensive interaction interface, accounting for their high affinity binding. Therefore, the endogenous lysoPS has occupied the receptor, leading to maximal activation and making it not respond to exogenous lysoPS in the cAMP assay. This also explains why high concentration of lysoPS is required to produce any effect in vivo.

Please cite several important and already published articles, which demonstrate that LysoPS is the endogenous ligand of GPR174 (PMID: 22983457).

Response: All papers including PMID: 22983457 that demonstrate lysoPS is an endogenous ligand have already been cited in the original manuscript.

In addition, the author classified these 12 caoGPCRs into four types, but without any specific criteria. It is not reasonable according to their sequence similarity as described by the author. The phylogenetic tree analysis may be one of the ways to reflect the sequence conservation. The authors can provide more information about the mechanism of caoGPCRs with high basal activity if they can determine the structures of GPR26 or GPR6. Moreover, could the author get any common features (or motifs) of caoGPCRs from the current structure or data analysis?

Response: Thanks for pointing out. We reclassify these caoGPCRs into three major groups based on sequence and structural similarity: I (GPR26, GPR78, GPR101, GPR161), II (GPR3, GPR6, GPR12), III (GPR21, GPR52). The receptors in the group I are closely related with prostanoid receptors. Structural predictions by Alphafold reveal that ECL2 in the group I form an antiparallel β -sheet, covering the ligand-binding pocket, which is observed in the structure of GPR161 determined in this study and prostanoid receptors. GPR3, GPR6 and GPR12 in the group II share about 60% sequence identity. Sequence similarity in the ligand-binding pocket is even more conserved among these receptors. Given their high sequence similarity, the receptors in the group II likely share similar mechanisms underlying the high basal activity. GPR21 shares 71% sequence identity with GPR52. Recent structural studies suggest that GPR21 and GPR52 can be self-activated by ECL2, which contributes to their high constitutive activity. The other three caoGPCRs are not classified due to their low similarity with the other receptors.

Sequence similarity is a reasonable criterion for classification. We agree that accurate classification needs more structural and functional studies, which will require a substantial amount of work and are beyond the scope of this study. Our future work will illustrate the molecular mechanisms of all caoGPCRs.

3. The author provided more information on GPR174 activation and identified the ligand LysoPS in the receptor. I suggest the author confirm the binding mode by other experiments (BRET assay or G protein dissociate assay).

Response: we have now further confirmed the binding mode by NanoBiT Gs recruitment assay (Supplementary Fig. 3d), which show consistent results with cAMP accumulation assay.

4. The author just tested the Gs signaling for these receptors. The author should measure the constitutive activities of other signaling pathways (Gi, Gq, G12/13, arrestin). Only determination of Gs activity for these receptors is an important flaw, because GPCRs were known to couple to 20 different G protein subtypes.

Response: In this study, we mainly focus on Gs signaling of oGPCRs. We will measure the constitutive activity of other signaling pathways for class A oGPCRs in the future work, but these are beyond the scope of this study.

5. Fig 1a shows the comparison of the signal intensity of D1R self-activation with different orphan receptors of class A receptor. This data is questionable. The receptors listed in supplementary table 1 showed varied cell surface expression levels, which will certainly lead to different downstream signaling pathway and different activities. The authors should ensure all receptors tested are expressed at similar cell surface expression levels by adjusting the plasmids amount transfected into the cells.

Response: It is very challenging to ensure all 81 receptors are expressed at similar levels. If we understand correctly, your concern is whether the extremely high basal activities of caoGPCRs are attributed to their expression levels, since the basal activity of GPCRs is proportional to the expression level.

Firstly, 9 of 12 caoGPCRs are expressed at a lower level compared to D1R, the other 3 caoGPCRs are expressed at comparable levels to D1R. Secondly, further efforts to increase the expression level of D1R has little effect on its basal activity, which is still remarkably lower than that of GPR174 even when expressed at an extremely low level (Figure below). Therefore, the higher constitutive activities of caoGPCRs are not attributed to their expression levels.

6. While the authors have provided PDB files for review, many figures and descriptions in the manuscript are not consistent with the provided PDB files. For example: (i) Figure 3c and lines 107-110 describe the phosphate group in lysoPS as covered by three spatially separated basic residues, the model of residues (R181.31, R1564.64, K2576.62) in GPR174-Gs don't fit with density.

Response: Thanks for pointing out. R18 is a little off. We have adjusted it.

(ii) the authors describe many salt bridge interactions in Figure 4b and lines 150-153 between glutamic acid or aspartic acid with Arginine or Lysine, but at least two residues (K298ECL3, E3067.35) lack the density of this part, the modeling is very ambiguous with the map density in this region;

Response: The EM densities for these residues are very poor likely due to their flexibility. I removed the side chains that show poor density in the PDB. Since the K298 is in close distance to D172, E306, and E293, it is very likely that they form salt bridge interactions. We now add “potentially” in the description.

(iii) In GPR174-Gs structure, there is almost no EM density around C11 and C16 of lysoPS. Figure 1c didn't reflect the fact and is misleading;

Response: This depends on what contour level is used. We use contour level 3.5 where noise is barely visible. We have added the contour level value in the figure legend.

(iv) In GPR61-Gs structure, the van der Waals shown in Figure 4f between the W199ECL2 and TM4, is also not supported by the model provided, with almost no density for W199 ECL2 of GPR61.

Response: We can observe partial density for the side chain of W199, which is attributed to the relatively low resolution of the extracellular side region, but the position should be accurate.

Moreover, there is a numbering of modeling errors in the PDB files provided for review (poor rotamers/poor fit to the experimental density maps) that need to be addressed. I would recommend the authors thoroughly check through the PDB files with the density maps. There are also many regions (outside of the regions discussed specifically in the manuscript) within the PDB files that are modeled differently in the 3 structures, which are not supported by differences in density in the cryo-EM maps. These are generally regions that have poor density, so modeling accurately is difficult, therefore I would recommend the authors either remove these regions from the models or if modeled, model these regions the same for all structures, unless there is sufficient density to identify clear differences between the different structures. These are key issues because the data don't support their scientific claims.

Response: We have fixed the poor rotamers/poor fit. We also removed the regions that show poor EM density.

7. In this article, the authors used the nanobit system to recombine the complex of GPR61–Gs, GPR161–Gs and GPR174–Gs, regardless of whether there are any physiological or functional supports for corresponding signaling complexes. The nanobit system is easy to artificially tether the complex even if they don't exist in physiological conditions. So, does this really exist under physiological conditions?

Response: We used the fusion protein strategy but not the NanoBiT system to tether GPCR and G proteins. The length of the linker between GPCR and G protein we used is much longer than the distance between the C-terminus of GPCR and N-terminus of G α . To our knowledge, if the two proteins cannot bind in physiological conditions, it is unlikely to force them to form a complex by introducing a linker. Our group including other groups have determined a number of GPCR-G protein complexes using the fusion protein strategy. All these structures can be validated by functional assays and are consistent with structures determined by other groups using different approaches. In our previous work (PMID: 35594396), we found that cleavage of the linker between galanin receptor and G protein did not change the overall structure of galanin and G protein complex. Moreover, GPR174 has been shown to signal via Gs in several studies.

Minor concerns:

1. The cAMP accumulation effect of R156A(4.64) was reduced to ~35%, and there was no significant increase under the treatment of a higher concentration of LysoPS, which is contradicted by what is described in the text.

Response: Some mutations such as Y99A reduce the potency (EC50) of lysoPS but not the maximal response (Emax). In contrast, mutations such as R156A reduced the Emax

but not the EC50. Therefore, endogenous lysoPS can still fully activate the R156A mutant.

2.As lysine has only one amino group, I am curious whether it can form two ionic bonds with E293 and E306 at the same time. (Line152)

Response: Ionic bond is the electrostatic attraction between oppositely charged ions. Therefore, it is possible that one lysine form two ionic bonds with E293 and E306 at the same time as long as they are close.

3.As is described in line157, "ECL2 in the structure of GPR61 and previously reported structures of GPR21, GPR52 and GPR12 are organized into a short loop structure (Fig. 4d)." The short loop structure of GPR21, GPR52 and GPR12 should also be shown, to present the similarities and differences with GPR61.

Response: Comparison of ECL2 in GPR21, GPR52 and GPR61 is added in supplementary Fig. 6b. ECL2 in GPR21 and GPR52 penetrates deeper in the orthosteric pocket than that in GPR61. The structure of GPR12 has not been released and therefore not included.

4.Q198 appears to interact with H307, however, the substitution of Q198 instead enhances the constitutive activity. It is necessary to include more discussion and come up with some reasonable explanations.

Response: We speculate that interaction between Q198 and H307 in TM6 may limit the outward movement of TM6. Therefore, abolishing their interaction may lower energy barrier of the active state and enhance the constitutive activity.

5.Control curves should be included (e.g., Fig3e and Fig6d). For example, Fig6d should include the curve of the Y99A single mutant.

Response: we have added the control curves.

6. In Figure 3c, the assumed LysoPS form extensive contacts with GPR174, but the authors only examined the contribution of R75, Y99, R156, and K257 to basal activity and signals, the other mutants are needed to test to illustrate a more detailed mechanism. Moreover, the authors must provide ELISA assay data or Western Blot data for mutants to ensure similar cell surface expression levels of WT and its mutants.

Response: We have included mutations in hydrophobic residues that surround the acyl chain of lysoPS. Western blot for mutants is also included (Fig 3d, 3e).

7.The GPR174-LysoPS-Gs structure has already been solved by He et al, Nat Commun, 2023, and the resolution is higher than that in this manuscript. But the authors did not include any discussions about this. Whether the structure models are similar or different? Is there any new insight?

Response: We had submitted our paper before that paper was published. Our structure is similar to theirs. We added discussion about that work in line 298-304.

8. The previous study (Journal of Neuroscience Research, 2009, Doi: 10.1002/JNR.21955) mentioned that the N-terminal has an important role in GPR61 constitutive activity. In the study, why only considered the role of ECL2 in its constitutive activity? The role of the N-terminal was not mentioned.

Response: N-terminus of GPR61 cannot be observed in our structure and therefore is not mentioned. The cited study suggests that the N-terminus of GPR61 is important for its membrane translocation.

9. Line 83, "We speculated the endogenous ligands might have occupied the receptor, leading to its maximal activation in the cAMP assay". According to the results shown in fig1c, the authors have the above conjecture that the occupation of pocket by endogenous ligand will preclude the activation of the receptor by additional ligand LysoPS. But how do the authors explain the decline in the back of the curve?

Response: Owing to its amphipathic characteristics, high concentration of lysoPS disrupts membrane structure and causes cell lysis, thereby leading to a reduced cAMP level. The decline of cAMP curve at high concentration of lysoPS is also observed in other caoGPCRs.

10. The density of ligands shown in Fig3b still has unassigned areas. Could the authors continue to work on the existing data to enhance the precision of ligands or consider other endogenous substances inherent in 293 cells?

Response: Sorry for the confusion. The grey area in Figure 3b shows the interior surface of GPR174 but not the EM density.

11. Line 114 mentioned that, as shown in Fig3e, GPR174 had a dose-dependent signal activation on LysoPS after the four amino acids (R752.60, Y993.33A, R1564.64A or K2576.62) were mutated into alanine. But why the signal data of WT is not shown in Fig3e at the same time as the control. Comparison could not be derived from two independent experiments.

Response: We have added signal data of WT in the Fig. 3e.

12. Fig4c and 4f clearly reflect the same type of experimental data, so it is recommended to use a combination of similar order when arranging the figure. For example, the position of "Empty" should be in the same position, to increase the beauty and rationality of the figure.

Response: Thanks for your suggestion. We have changed the position.

13. Does the C-terminal mini-Gs399 fusion protein affect the conformation of C-terminus of GPR174, GPR161 or GPR61? The authors should clarify by measuring their activities.

Response: The fusion linker between the C-terminus of GPCR and the N-terminus of mini-Gs is about 50 aa, which is about 150 Å. Their actual distances are 56.0 Å, 35.4 Å and 42.6 Å for GPR174-mini-Gs, GPR161-mini-Gs and GPR61-mini-Gs structure respectively.

Therefore, the linker is long enough to cover the spatial distance.

As mentioned above, in our previous work (PMID: 35594396), we found that cleavage of the linker between galanin receptor and G protein did not change the overall structure of galanin and G protein complex. We can combine the EM dataset collected from the galanin-G fusion protein complex and the non-fusion complex where the linker is cleaved for structure determination.

14. The order of the histograms of fig4c and 4f should be consistent, including the order of empty, WT and mutations.

Response: We have changed their order.

15. The author used a large number of highly saturated colors for the diagrams involving structural comparisons, which could not highlight the meaning he wanted to express.

Response: Thanks for your suggestion. We have revised some figures.

16. Whether the GPR174–mini–Gas fusion protein affect the conformation of C-terminus in mini–Gas? The authors should perform molecular dynamic simulation to confirm the distorted “hook” conformation.

Response: Please refer to response to minor point 13. The distorted hook conformation is also observed in the structure of GPR174-Gs published by other group using different strategy. The binding interface can be validated by mutagenesis experiments.

Reviewer #2 (Remarks to the Author):

This study determined 3 active structures of GPR61, GPR161 and GPR174, which are orphan GPCRs with high basal activity. In addition, they also identified the endogenous ligand of GPR174 and investigated the self-activation mechanism of GPR61 and GPR161. Most importantly, they revealed a non-canonical Gs coupling mode. It is very interesting and will provide useful information for the structural studies of orphan GPCRs. I have some concerns as below before the manuscript will be considered by publication in Nature Communications.

The major ones:

1) For GPR174, only 4 mutants were tested but there are a lot of interactions between GPR174 and lysoPS. Did you test other mutants, for example, the residues in the deeper hydrophobic pocket?

Response: Thanks for your advice! We have tested more mutations in the deep hydrophobic pocket using both cAMP accumulation assay (Fig. 3d, 3e) and NanoBIT Gs recruitment assay (Supplementary Fig. 3d).

2) The part of self-activation of GPR61 and GPR161 is weak. There are some other GPCRs which are activated by ECL2, it is better that add sequence and structure comparisons of

GPR61, GPR161, GPR52, GPR21 and so on.

Besides, all the ECL2 mutations of GPR61 didn't kill the basal activity, which indicate that ECL2 may not be the only activation factor. What if replace the whole ECL2 with GSSGSS linker or deletion of the ECL2?

Response: Thanks for your suggestion. Comparison of structures of GPR61, GPR52 and GPR21 is added in Supplementary Fig. 6b. Instead of forming β hairpin structure in GPR161, ECL2 in the structure of GPR61 and previously reported structures of GPR21, GPR52 and GPR12 are organized into a short loop structure. ECL2 in GPR61 shows shallower penetration in the orthosteric pocket compared to that in GPR21 and GPR52.

Replacement of ECL2 (SLQW) with GGGG linker in GPR61 almost totally impairs the basal activity. As discussed in the paper, although mutations of ECL2 impair the basal activity of these receptors, we cannot exclude the possibility that endogenous ligands may contribute the high basal activity since ECL2 may be involved in ligand binding.

3) In the Gs binding mode part, it was mentioned that L5.65 is the key residue for the non-canonical Gs coupling. So, what is the result of L5.65 mutation? Looks like there is no experimental results for this mutation. The fig 6b is just a structural comparison and it doesn't match the statement in line 191.

Response: We have added L^{5.65} mutagenesis data. In contrast to D1R that prefers a small hydrophobic residue (A or V) at the position 5.65, mutation of L212^{5.65} in alanine remarkably reduces the Gs coupling efficiency either in the context of GPR174WT or GPR174/Y99A mutant (Fig. 6c, 6d). We have removed Fig. 6b reference.

Small concerns:

1) figure 3b, the grey density in the pocket is obviously larger than the lysoPS and it is different from fig 2c. Is it also including part of the receptor side chain density or is there any extra density in the pocket except lysoPS?

Response: Sorry for the confusion. The grey density is not cryo-EM density map for lysoPS in figure 3b. It is the interior surface of GPR174 model. We have modified the text to improve clarity.

2) figure 3c, I noticed that Y79 is labelled but didn't mentioned in the paper, what kind of interaction of this residue?

Response: Y79 forms a hydrogen bond with the amino group in the serine head of lysoPS. We added this in the text.

3) line 124, the current data in the paper didn't test G12/G13, so maybe remove this statement.

Response: We removed this statement.

4) line 178, what is the outward movement distance of TM6 in active beta2AR and μ opioid receptor? It should be a clear statement including the exact number.

Response: Compared with inactive β 2AR, TM6 in GPR174 moves outward by only 2 Å, which is smaller than the outward displacement of TM6 in the active β 2AR (14 Å) and the active μ opioid receptor that is bound to Gai (6 Å)

Reviewer #3 (Remarks to the Author):

Nie Y. et al reported cryo-EM structures of GPR61-Gs, GPR161-Gs, and GPR174-Gs signalling complexes without exogenous ligands. By analyzing these structures, they discussed the structural features of constitutively active orphan GPCRs (caoGPCRs), and found out that these receptors can be fully activated by endogenous ligands (e.g., GPR174 showing high affinity to endogenous LysoPS), or by occupation of ECL2 in the orthosteric site in the absence of endogenous ligand (e.g., GPR161 and GPR61). The coupling mode of Gs protein was also discussed, and their results suggested a possibility that some of GPCRs (such as GPR174) are capable of coupling with other G protein subtypes rather than Gs, concluding that a non-canonical Gs coupling mode was taken. The topic is intriguing; however the contents are scientifically ambiguous.

Major Concerns:

1) About the definition of “constitutive active”: usually “constitutive active” means Constitutive (basal) activity is defined as “ligand independent activity, resulting in the production of a second messenger in the absence of an agonist” (Scientific Reports volume 6, Article number: 38564 (2016)). However, the title and the main topic of this paper described “ligand-dependent” constitutive activation of GPR174. This is a new concept if the observation is correct. In order to avoid the confusion, the authors may use another terminology.

Response: Thanks for your suggestion! We change the title to “Specific binding of GPR174 by endogenous lysophosphatidylserine leads to high constitutive activity”. The constitutive activity here means the activity of GPR174 without exogenous ligand treatment. We know it is not precise, but cannot come up with a more reasonable term.

2) The authors mentioned endogenous lysophosphatidylserine (lysoPS) always binds to GPR174 and GPR174 was activated. The authors proposed two factors: abundance of lysoPS, and high affinity of lysoPS to GPR174. So the authors will show experimentally binding constant of lysoPS to GPR174.

Response: Thanks for your suggestion! We would like to measure the binding affinity between GPR174 and lysoPS. We have tried different methods such as microscale thermophoresis (MST) or isothermal titration calorimetry (ITC), but both failed. We think this is because the purified GPR174 may have already been bound by the endogenous lysoPS, making it difficult to measure the binding affinity.

In MST, lysoPS cause the GPR174's curve (Fig. a) to decline at micromolar lysoPS concentration, since high concentration of lysoPS may disrupt the receptor micelle. The same decline was observed in Fig. b, in which lysoPS was titrated to D1R as negative control. In ITC, heat released upon binding is too small to measure the binding affinity.

Instead, we sought to measure the potency of lysoPS for WT using the NanoBiT G_s recruitment assay which has a lower amplification than the cAMP assay (Supplementary Fig. 3d). For D1R, the potencies of dopamine obtained from the NanoBiT assay and cAMP assay are 2.1 μM and 3.6 nM (about 1000-fold difference), respectively (Supplementary Figs. 3e, 3f). As expected, lysoPS can activate GPR174 WT in a dose-dependent manner with a potency of 155.7 nM using the NanoBiT G_s recruitment assay (Supplementary Table 3). For the Y99^{3.33}A mutant, the potency is reduced by about 7-fold compared to GPR174 WT in the NanoBiT assay (Supplementary Table 3). Therefore, we speculate that the potency of lysoPS for WT using the cAMP assay could be in a single-digit nanomolar range (a potency of 71.6 nM for Y99^{3.33}A in the cAMP assay and about 7-fold weaker than WT).

3) The authors try to exclude lipidic materials including lysoPS by addition of serum albumin and do the cAMP assay.

Response: We have added serum albumin to exclude lipidic materials and did the cAMP assay. The basal activity of GPR174 is reduced only when high concentration of serum albumin (1%) is added, and D1R has a similar result, probably due to the toxicity of high concentration of BSA to cells or non-specific binding of BSA to luciferin. We think it is

very challenging for BSA to compete lysoPS from GPR174 given their high binding

affinity.

4) The authors will examine other cell lines (at least two, additionally) to check the reproducibility of the phenomenon. If there is not reproducible, the present cell line will represent a pathogenic constitutive active state of GPR174.

Response: We have now examined two cell lines (CHO and HeLa cell. As expected, GPR174 elevates cellular cAMP level in both cell line compared to DRD1.

5) In the case of GPR161, the binding between ECL2 and the orthosteric pocket was discussed by mutations of residues on ECL2. The mutations of the counterpart (e.g., R184ECL2 that binds to E170ECL2, E2936.58 and E3067.35 that are interacting with K175ECL2, and the hydrophobic residues interacting with M177ECL2) should also be carried out to confirm their hypothesis.

Response: We have added more mutations.

6) Which isoform of LysoPS (acyl chain length) was used in the author's cAMP accumulation assay? Please specify it.

Response: We use lysoPS 18:1. This is now stated in the methods.

7) Line 226. "Mutation of F124^{34.51} ..." However, the interaction between F124 in GPR174 and Gα was not specified in Figure 6c. Please specify.

Response: The residue F124^{34.51} is specified in Supplementary Fig 6d.

8) This year, Liang J. et al also reported the cryo-EM structure of GPR174 (Nature Communications volume 14, Article number: 1012 (2023)), which showed similar ligand binding mode and Gs binding mode and GPR174. So what is the new finding in this paper?

Response: Our manuscript was submitted before the cited study was published. Previous studies have shown that very high concentrations of lysoPS are required to promote GPR174-mediated suppression of T cell proliferation. Moreover, our study reveals that lysoPS fails to stimulate GPR174-mediated cAMP accumulation in the Glosensor assay. Liang J et al., also found that lysoPS cannot stimulate GPR174 mediated Gs dissociation using the NanoBiT dissociation assay. Therefore, a key question is why lysoPS cannot activate GPR174 using these assays if lysoPS is the endogenous ligand. Our study answers this question. We found that lysoPS among all endogenous lipids is specifically copurified with GPR174, indicating that lysoPS is the most potent ligand among all lipids for GPR174. The endogenous lysoPS has occupied the receptor, leading to maximal activation and making it not respond to exogenous lysoPS in the cAMP assay and NanoBiT dissociation assay. This also explains why high concentration of lysoPS is required to produce any effect in vivo.

9) In Liang's result (Nature Communications volume 14, Article number: 1012 (2023)), Y99^{3.33} interacts with carboxylate of serine head instead of sn-2/sn-3 hydroxy group (sn-2 hydroxy group interacts with Y246 instead). Another medicinal chemistry study by Ikubo M. et al (J. Med. Chem., 2015, 58 (10), pp 4204-4219) suggested the critical role of sn-2 hydroxy group in GPR174 activity. How can we explain the agonistic activity of Y99A GPR174 mutant induced with respect to exogenous LysoPS?

Response: Different conformations of Y99 and Y246 from Liang's structure may be due to the presence of cholesterol in our structure (Figure below). As discussed above, the endogenous lysoPS has occupied GPR174, leading to maximal activation and making it not respond to the exogenous lysoPS in the cAMP assay. However, the endogenous lysoPS can not fully activate Y99A mutant because of its reduced binding affinity and therefore activates Y99A mutant in a dose dependent manner in the cAMP assay. Moreover, we measure the potency of lysoPS for WT using the NanoBiT G_s recruitment assay which has a lower amplification than the cAMP assay (Supplementary Fig. 3d). As expected, lysoPS can activate GPR174 WT in a dose-dependent manner with a potency of 155.7 nM using the NanoBiT G_s recruitment assay (Supplementary Table 3). For the Y99^{3.33}A mutant, the potency is reduced by about 7-fold compared to GPR174 WT in the NanoBiT assay (Supplementary Table 3). These results indicated the important role of Y99 in lysoPS binding.

10) The authors need to more discuss the relationship between the non-canonical binding of Gs protein and constitutive active state.

Response: We think that the non-canonical Gs coupling mode is associated with promiscuous G protein coupling but not high constitutive activity. GPCRs such as GPR174, and GPR21, GPR52, CCK1R and NK1R that adopts non-canonical Gs coupling mode can couple to other G protein subtypes in addition to Gs. Most caoGPCRs such as GPR3, GPR6, GPR12, GPR61, GPR161, and GPR101 adopt canonical Gs coupling mode. The high constitutive activities of caoGPCRs are mainly attributed to the endogenous ligands or maybe self-activation by ECL2. The discussion is included in the paper.

REVIEWER COMMENTS

Reviewer #1 (Remarks to the Author):

Overall, the author has provided responses to most of my concern, but unfortunately, the majority of the answers are merely explanations without further experimental or data support. For me, these replies are not very convincing, and I suggest that the author further revise and clarify in accordance with the previous requests.

Reviewer #2 (Remarks to the Author):

In this study, the authors determined the Cryo-EM structures of three orphan receptors without exogenous ligands and found an endogenous ligand on GPR174. In the revised version, the authors did substantial experiments (NanoBiT Gs recruitment assay, etc.) suggested by the reviewers and fully addressed all the points raised by the reviewers. I believe the revision should meet all the reviewers and be suitable for publication in Nature Communications.

Reviewer #3 (Remarks to the Author):

This is the revised manuscript by Zheng et al which I reviewed. The authors carefully added new experimental results and explanation and revised the title which minimizes misleading for the audience. The authors addressed to all queries which I raised in my first review. Therefore, I recommend this work to publish Nature Communications.

Editorial note: Final response to referee concerns.

Reviewer #1 (Remarks to the Author):

This manuscript by Nie et al. focused on a group of orphan class A/rhodopsin-like GPCRs, including GPR61, GPR161, and GPR174, with high constitutive activity. Using single-particle cryo-EM, the authors determined the structures of GPR61-Gs, GPR161-Gs, and GPR174-Gs complexes without exogenous ligands. The structures of GPR161 and GPR61 reveal that the second extracellular loop (ECL2) penetrates into the orthosteric pocket, and the GPR174 adopts a non-canonical Gs coupling mode. In addition, the authors claimed that the GPR174 is an orphan receptor, and LysoPS is its endogenous ligand. Overall, I think this manuscript has several serious flaws to be addressed before publication. I therefore recommend for a major revision.

Major concerns:

1. The authors determined the structure of GPR174 and modeled LysoPS with 15 carbons in the acyl into the unassigned density. In my opinion, Figure 2C is misleading, since it does not match the lipidomics results. The lipidomics indicated that the isoforms from 16:0 to 20:1 are determined from Mass Spectrum analysis. The density of LysoPS in the current structure is unconvincing, the author should further confirm the binding pose of LysoPS in GPR174 by determining the structure of GPR174 with specific LysoPS molecule (LysoPS 18:1) or perform computational simulations.

Response: Thanks for bringing this up! It is not surprising we can only observe EM density for lysoPS with 15 carbons even though 16:0 to 20:1 are highly enriched. We think that the carbons that are absent in our structure are likely to be very flexible due to the lack of contact with the receptor. We confirmed its binding pose using molecular dynamics (MD) simulations as you suggested. LysoPS stably associates with GPR174 during the course of 100-ns MD (Supplementary Fig. 3b). The acyl chain particularly the first two carbons displayed more flexibility than the polar group, explaining the absence of the terminal carbons in the structure. We have added this part in the paper.

2. In this manuscript, the authors would like to understand the activation mechanism of orphan receptors. However, LysoPS was already identified as an endogenous ligand for GPR174 in previous studies, this receptor doesn't belong to an orphan receptor.

Response: Although endogenous ligands have been shown to activate some orphan GPCRs, they are still classified as orphan in GPCRdb. One reason is that there is lack of sufficient evidence. For GPR174, in addition to lysoPS, other group has reported CCL21 is a potential ligand (PMID: 31875850). Dr. Jason Cyster group found that high concentrations of lysoPS are required to promote GPR174-mediated suppression of T cell proliferation, arguing whether other endogenous ligands of GPR174 with high potency exist. Moreover, our work showed that lysoPS fails to stimulate cAMP accumulation in cells expressing GPR174 using the Glosensor cAMP assay. All these issues can be addressed in our study. We found that lysoPS among all endogenous lipids is specifically copurified with GPR174, supporting that lysoPS is the most potent ligand among all lipids for GPR174. The structure further shows perfect shape complementarity and extensive interaction interface, accounting for their high affinity binding. Therefore, the endogenous lysoPS has occupied the receptor, leading to maximal activation and making it not respond to exogenous lysoPS in the cAMP assay. This also explains why high concentration of lysoPS is required to produce any effect in vivo.

Please cite several important and already published articles, which demonstrate that LysoPS is the endogenous ligand of GPR174 (PMID: 22983457).

Response: All papers including PMID: 22983457 that demonstrate lysoPS is an endogenous ligand have already been cited in the original manuscript.

In addition, the author classified these 12 caoGPCRs into four types, but without any specific criteria. It is not reasonable according to their sequence similarity as described by the author. The phylogenetic tree analysis may be one of the ways to reflect the sequence conservation. The authors can provide more information about the mechanism of caoGPCRs with high basal activity if they can determine the structures of GPR26 or GPR6. Moreover, could the author get any common features (or motifs) of caoGPCRs from the current structure or data analysis?

Response: Thanks for pointing out. We reclassify these caoGPCRs into three major groups based on sequence and structural similarity: I (GPR26, GPR78, GPR101, GPR161), II (GPR3, GPR6, GPR12), III (GPR21, GPR52). The receptors in the group I are closely related with prostanoid receptors. Structural predictions by AlphaFold reveal that ECL2 in the group I form an antiparallel β -sheet, covering the ligand-binding pocket, which is observed in the structure of GPR161 determined in this study and prostanoid receptors. GPR3, GPR6 and GPR12 in the group II share about 60% sequence identity. Sequence similarity in the ligand-binding pocket is even more conserved among these receptors. Given their high sequence similarity, the receptors in the group II likely share similar mechanisms underlying the high basal activity. GPR21 shares 71% sequence identity with GPR52. Recent structural studies suggest that GPR21 and GPR52 can be self-activated by ECL2, which contributes to their high constitutive activity. The other three caoGPCRs are not classified due to their low similarity with the other receptors. We have

modified this part in the paper.

Sequence similarity is a reasonable criterion for classification. We agree that accurate classification needs more structural and functional studies, which will require a substantial amount of work and are beyond the scope of this study. Our future work will illustrate the molecular mechanisms of all caoGPCRs.

3. The author provided more information on GPR174 activation and identified the ligand LysoPS in the receptor. I suggest the author confirm the binding mode by other experiments (BRET assay or G protein dissociate assay).

Response: we have now further confirmed the binding mode by NanoBiT Gs recruitment assay (**Fig. 3e**), which show consistent results with cAMP accumulation assay.

4. The author just tested the Gs signaling for these receptors. The author should measure the constitutive activities of other signaling pathways (Gi, Gq, G12/13, arrestin). Only determination of Gs activity for these receptors is an important flaw, because GPCRs were known to couple to 20 different G protein subtypes.

Response: In this study, we mainly focus on Gs signaling of oGPCRs. We will measure the constitutive activity of other signaling pathways for class A oGPCRs in the future work, but these are beyond the scope of this study.

5. Fig 1a shows the comparison of the signal intensity of D1R self-activation with different orphan receptors of class A receptor. This data is questionable. The receptors listed in supplementary table 1 showed varied cell surface expression levels, which will certainly lead to different downstream signaling pathway and different activities. The authors should ensure all receptors tested are expressed at similar cell surface expression levels by adjusting the plasmids amount transfected into the cells.

Response: It is very challenging to ensure all 81 receptors are expressed at similar levels. If we understand correctly, your concern is whether the extremely high basal activities of caoGPCRs are attributed to their expression levels, since the basal activity of GPCRs is proportional to the expression level.

Firstly, 9 of 12 caoGPCRs are expressed at a lower level compared to D1R, the other 3 caoGPCRs are expressed at comparable levels to D1R. Secondly, further efforts to increase the expression level of D1R has little effect on its basal activity, which is still remarkably lower than that of GPR174 even when expressed at an extremely low level (Supplementary Figs. 1a and 1b). Therefore, the higher constitutive activities of caoGPCRs are not attributed to their expression levels.

6. While the authors have provided PDB files for review, many figures and descriptions in the manuscript are not consistent with the provided PDB files. For example: (i) Figure 3c and lines 107-110 describe the phosphate group in lysoPS as covered by three spatially separated basic residues, the model of residues (R18^{1.31}, R156^{4.64}, K257^{6.62}) in

GPR174-Gs don't fit with density.

Response: Thanks for pointing out. R18 is a little off. We have adjusted it. **We have included EM maps for residues involved in binding lysoPS in Supplementary Fig. 2f.**

(ii) the authors describe many salt bridge interactions in Figure 4b and lines 150-153 between glutamic acid or aspartic acid with Arginine or Lysine, but at least two residues (K298^{ECL3}, E306^{7.35}) lack the density of this part, the modeling is very ambiguous with the map density in this region;

Response: The EM densities for these residues are very poor likely due to their flexibility. I removed the side chains that show poor density in the PDB. Since the K298 is in close distance to D172, E306, and E293, it is very likely that they form salt bridge interactions. We now add "potentially" in the description.

(iii) In GPR174-Gs structure, there is almost no EM density around C11 and C16 of lysoPS. Figure 1c didn't reflect the fact and is misleading;

Response: This depends on what contour level is used. We use contour level 3.5σ where noise is barely visible. We have added the contour level value in the figure legend.

(iv) In GPR61-Gs structure, the van der Waals shown in Figure 4f between the W199ECL2 and TM4, is also not supported by the model provided, with almost no density for W199 ECL2 of GPR61.

Response: We can observe partial density for the side chain of W199, which is attributed to the relatively low resolution of the extracellular side region, but the position should be accurate. We have included EM maps for W199 in **Supplementary Fig. 5f.**

Moreover, there is a numbering of modeling errors in the PDB files provided for review (poor rotamers/poor fit to the experimental density maps) that need to be addressed. I would recommend the authors thoroughly check through the PDB files with the density maps. There are also many regions (outside of the regions discussed specifically in the manuscript) within the PDB files that are modeled differently in the 3 structures, which are not supported by differences in density in the cryo-EM maps. These are generally regions that have poor density, so modeling accurately is difficult, therefore I would recommend the authors either remove these regions from the models or if modeled, model these regions the same for all structures, unless there is sufficient density to identify clear differences between the different structures. These are key issues because the data don't support their scientific claims.

Response: We have fixed the poor rotamers/poor fit. We also removed the regions that show poor EM density.

7. In this article, the authors used the nanobit system to recombine the complex of GPR61-Gs, GPR161-Gs and GPR174-Gs, regardless of whether there are any physiological or functional supports for corresponding signaling complexes. The nanobit system is easy to artificially tether the complex even if they don't exist in physiological conditions. So, does this really exist under physiological conditions?

Response: We used the fusion protein strategy but not the NanoBiT system to tether

GPCR and G proteins. The length of the linker between GPCR and G protein we used is much longer than the distance between the C-terminus of GPCR and N-terminus of G α . To our knowledge, if the two proteins cannot bind in physiological conditions, it is unlikely to force them to form a complex by introducing a linker. Our group including other groups have determined a number of GPCR-G protein complexes using the fusion protein strategy. All these structures can be validated by functional assays and are consistent with structures determined by other groups using different approaches. In our previous work (PMID: 35594396), we found that cleavage of the linker between galanin receptor and G protein did not change the overall structure of galanin and G protein complex. Moreover, GPR174 has been shown to signal via Gs in several studies.

Minor concerns:

1. The cAMP accumulation effect of R156A(4.64) was reduced to ~35%, and there was no significant increase under the treatment of a higher concentration of LysoPS, which is contradicted by what is described in the text.

Response: Some mutations such as Y99A reduce the potency (EC₅₀) of lysoPS but not the maximal response (E_{max}). In contrast, mutations such as R156A reduced the E_{max} but not the EC₅₀. Therefore, endogenous lysoPS can still fully activate the R156A mutant.

2. As lysine has only one amino group, I am curious whether it can form two ionic bonds with E293 and E306 at the same time. (Line152)

Response: Ionic bond is the electrostatic attraction between oppositely charged ions. Therefore, it is possible that one lysine form two ionic bonds with E293 and E306 at the same time as long as they are close.

3.As is described in line157, "ECL2 in the structure of GPR61 and previously reported structures of GPR21, GPR52 and GPR12 are organized into a short loop structure (Fig. 4d)." The short loop structure of GPR21, GPR52 and GPR12 should also be shown, to present the similarities and differences with GPR61.

Response: Comparison of ECL2 in GPR21, GPR52 and GPR61 is added in supplementary Fig. 6b. ECL2 in GPR21 and GPR52 penetrates deeper in the orthosteric pocket than that in GPR61. The structure of GPR12 has not been released and therefore not included.

4. Q198 appears to interact with H307, however, the substitution of Q198 instead enhances the constitutive activity. It is necessary to include more discussion and come up with some reasonable explanations.

Response: We speculate that interaction between Q198 and H307 in TM6 may limit the outward movement of TM6. Therefore, abolishing their interaction may lower energy barrier of the active state and enhance the constitutive activity.

5. Control curves should be included (e.g., Fig3e and Fig6d). For example, Fig6d should include the curve of the Y99A single mutant.

Response: we have added the control curves.

6. In Figure 3c, the assumed LysoPS form extensive contacts with GPR174, but the authors only examined the contribution of R75, Y99, R156, and K257 to basal activity and signals, the other mutants are needed to test to illustrate a more detailed mechanism. Moreover, the authors must provide ELISA assay data or Western Blot data for mutants to ensure similar cell surface expression levels of WT and its mutants.

Response: We have included mutations in hydrophobic residues that surround the acyl chain of lysoPS. Western blot for mutants is also included (Fig 3d, 3e).

7. The GPR174-LysoPS-Gs structure has already been solved by He et al, Nat Commun, 2023, and the resolution is higher than that in this manuscript. But the authors did not include any discussions about this. Whether the structure models are similar or different? Is there any new insight?

Response: We had submitted our paper before that paper was published. Our structure is similar to theirs. We added discussion about that work in line 298-304.

8. The previous study (Journal of Neuroscience Research, 2009, Doi: 10.1002/JNR.21955) mentioned that the N-terminal has an important role in GPR61 constitutive activity. In the study, why only considered the role of ECL2 in its constitutive activity? The role of the N-terminal was not mentioned.

Response: N-terminus of GPR61 cannot be observed in our structure and therefore is not mentioned. The cited study suggests that the N-terminus of GPR61 is important for its membrane translocation.

9. Line 83, "We speculated the endogenous ligands might have occupied the receptor, leading to its maximal activation in the cAMP assay". According to the results shown in Fig1c, the authors have the above conjecture that the occupation of pocket by endogenous ligand will preclude the activation of the receptor by additional ligand LysoPS. But how do the authors explain the decline in the back of the curve?

Response: Owing to its amphipathic characteristics, high concentration of lysoPS disrupts membrane structure and causes cell lysis, thereby leading to a reduced cAMP level. The decline of cAMP curve at high concentration of lysoPS is also observed in other GPCRs.

10. The density of ligands shown in Fig3b still has unassigned areas. Could the authors continue to work on the existing data to enhance the precision of ligands or consider other endogenous substances inherent in 293 cells?

Response: Sorry for the confusion. The grey area in Figure 3b shows the interior surface of GPR174 but not the EM density.

11. Line 114 mentioned that, as shown in Fig3e, GPR174 had a dose-dependent signal

activation on LysoPS after the four amino acids (R752.60, Y993.33A , R1564.64A or K2576.62) were mutated into alanine. But why the signal data of WT is not shown in Fig3e at the same time as the control. Comparison could not be derived from two independent experiments.

Response: We have added signal data of WT in the Fig. 3e.

12. Fig4c and 4f clearly reflect the same type of experimental data, so it is recommended to use a combination of similar order when arranging the figure. For example, the position of "Empty" should be in the same position, to increase the beauty and rationality of the figure.

Response: Thanks for your suggestion. We have changed the position.

13. Does the C-terminal mini-Gs399 fusion protein affect the conformation of C-terminus of GPR174, GPR161 or GPR61? The authors should clarify by measuring their activities.

Response: The fusion linker between the C-terminus of GPCR and the N-terminus of mini-Gs is about 50 aa, which is about 150 Å. Their actual distances are 56.0 Å, 35.4 Å and 42.6 Å for GPR174-mini-Gs, GPR161-mini-Gs and GPR61-mini-Gs structure respectively. Therefore, the linker is long enough to cover the spatial distance.

As mentioned above, in our previous work (PMID: 35594396), we found that cleavage of the linker between galanin receptor and G protein did not change the overall structure of galanin and G protein complex. We can combine the EM dataset collected from the galanin-G fusion protein complex and the non-fusion complex where the linker is cleaved for structure determination.

14. The order of the histograms of fig4c and 4f should be consistent, including the order of empty, WT and mutations.

Response: We have changed their order.

15. The author used a large number of highly saturated colors for the diagrams involving structural comparisons, which could not highlight the meaning he wanted to express.

Response: Thanks for your suggestion. We have revised some figures.

16. Whether the GPR174-mini-Gs fusion protein affect the conformation of C-terminus in mini-Gs? The authors should perform molecular dynamic simulation to confirm the distorted "hook" conformation.

Response: Please refer to response to minor point 13. The distorted hook conformation is also observed in the structure of GPR174-Gs published by other group using different strategy. The binding interface can be validated by mutagenesis experiments.